# *Yersinia* actively downregulates type III secretion and adhesion at higher cell densities

**Francesca Ermoli[1], Gabriele Malengo[2,3], Christoph Spahn[4,5], Corentin Brianceau[6], Timo Glatter[7], Andreas Diepold** [1,3,6]*

**1** Department of Ecophysiology, Max Planck Institute for Terrestrial Microbiology, Marburg, Germany,
**2** Core Facility for Flow Cytometry and Imaging, Max Planck Institute for Terrestrial Microbiology, Marburg, Germany, **3** Center for Synthetic Microbiology (SYNMIKRO), Marburg, Germany, **4** Department of Natural Products in Organismic Interactions, Max Planck Institute for Terrestrial Microbiology, Marburg, Germany, **5** Rudolf Virchow Center for Integrative and Translational Bioimaging, Julius-Maximilians-University Würzburg, Würzburg, Germany, **6** Department of Applied Biology, Institute of Applied Biosciences, Karlsruhe Institute of Technology (KIT), Karlsruhe, Germany, **7** Core Facility for Mass Spectrometry and Proteomics, Max Planck Institute for Terrestrial Microbiology, Marburg, Germany

* andreas.diepold@kit.edu

## Abstract

The T3SS injectisome is used by Gram-negative bacteria, including important pathogens, to manipulate eukaryotic target cells by injecting effector proteins. While in some bacterial species, T3SS-negative bacteria benefit from the activity of their T3SS-positive siblings, the T3SS model organism *Yersinia enterocolitica* was thought to uniformly express and assemble injectisomes. In this study, we found that *Yersinia* actively suppress T3SS expression, assembly and activity at higher cell densities, such as inside microcolonies. This effect is highly specific to the T3SS, reversible, and distinct from stationary phase adaptation. It is conferred by the main T3SS transcription factor VirF, which is downregulated at higher densities and whose *in trans* expression restores T3SS activity. The concomitant downregulation of the VirF-dependent adhesin YadA led to a drastic reduction in bacterial cell adhesion. We propose that this active suppression of T3SS secretion and cell attachment at higher local bacterial densities promotes a switch during *Yersinia* infection from a T3SS-active colonization stage to a bacterial replication and dissemination phase.

## Author summary

Bacteria can use the type III secretion system (T3SS), a molecular syringe-like device, to manipulate host cells by injecting effector proteins. *Yersinia enterocolitica*, a pathogenic bacterium related to *Yersinia pestis* – the causative agent of the black death – uses its T3SS to evade the immune system, allowing the bacteria to form microcolonies in the human body. However, T3SS activity comes at a cost, as bacteria that use the system stop growing and dividing. In this

**Data availability statement:** All relevant data is available in the manuscript, the supporting information, or the linked database entries. The mass spectrometry proteomics data have been deposited to the ProteomeXchange Consortium via the PRIDE partner repository with the dataset identifier PXD052013.

**Funding:** This work was supported by Max-Planck-Gesellschaft (to AD). The funders had no role in study design, data collection and analysis, decision to publish, or preparation of the manuscript.

**Competing interests:** The authors have declared that no competing interests exist.

study, we discovered that *Yersinia* balances the benefits and costs of the T3SS by switching off the system at higher cell densities, such as those found within microcolonies. This switch is mediated by regulatory RNAs that, at high density, block the production of a key protein required for the production of the T3SS, its injected effectors, and an adhesive protein. This regulatory mechanism enables the bacteria to transition from colonizing host tissue to replicating and spreading.

## Introduction

To manipulate eukaryotic cells, many Gram-negative bacteria, including important human pathogens, use a type III secretion system, which allows to inject effector proteins into target cells in a one-step mechanism [1–4]. These effector proteins can exert a wide variety of functions that facilitate survival and proliferation of the respective bacteria according to their infection route and lifestyle [5–7]. In contrast, the T3SS machinery, also known as injectisome, is highly conserved. It consists of a set of ring structures that anchor the injectisome in the bacterial membranes and surround an export apparatus in the inner membrane (IM), thought to act as a gating structure [8,9]. A hollow needle structure forms a conduit for the effectors, which is connected to a translocon pore in the target cell membrane (Fig 1a). At the cytosolic interface of the injectisome, a set of conserved cytosolic components form six pod structures [10–14] that shuttle effectors between the injectisome and the cytosol in *Yersinia enterocolitica* [15–17].

Expression, assembly and activity of the T3SS are tightly regulated. Often, expression and assembly only occur within the host organism. In *Yersinia*, this is based on an RNA thermometer structure in the transcript of the main transcriptional activator of the T3SS, called VirF in *Y. enterocolitica* and LcrF in *Y. pseudotuberculosis* and *Y. pestis* [18,19]. While expression of VirF is sufficient for the formation of injectisomes, a second signal, host cell contact, is required to activate secretion. This activation, which can be mimicked by $Ca^{2+}$ depletion or absence of the gatekeeper protein SctW in *Yersinia* [20,21], also removes the secreted posttranscriptional repressors YscM1/2 (LcrQ in *Y. pseudotuberculosis*) [22,23], leading to an upregulated production of the structural components and especially effectors of the system [23–25]. Recent studies in *Y. pseudotuberculosis* revealed additional layers of regulation (reviewed in [26]), such as the CsrABC system, where CsrA, which positively regulates T3SS expression by stabilizing the *virF* transcript, is sequestered by the two regulatory RNAs CsrB and CsrC [27].

This tight regulation of T3SS expression and activity may be required because of a phenomenon called secretion-associated growth inhibition (SAGI), in which secreting bacteria greatly reduce their growth and division. It was this striking phenotype that initially led to the discovery of the T3SS, when researchers noted a specific growth inhibition of virulent *Yersinia* (now known to be equivalent to the presence of the T3SS) at 37°C (T3SS expression) in the absence of $Ca^{2+}$ (T3SS activation) [28–33]. Besides *Yersinia*, SAGI has been shown to occur in two other main T3SS

model organisms, *Salmonella enterica* and *Shigella flexneri* [34–36]. However, it is still unclear if SAGI is a mere consequence of the energy expenditure for synthesis and export of the T3SS machinery and effectors, or whether it is an actively regulated phenotype, as suggested by the fact that growth and division are reinitiated very quickly, once secretion is stopped [37].

Given that the propagation of actively secreting bacteria is impeded, a differential expression of the T3SS within the population is conceivably beneficial in some situations. Indeed, it was shown that the SPI-1 T3SS, used by *Salmonella enterica* to induce a strong inflammation in the host gut that clears competing bacteria, is only active in a subset of bacteria, allowing the T3SS-inactive subpopulation to rapidly populate the infection niche [34,38]. Similarly, *Pseudomonas aeruginosa* expresses its T3SS in a bistable way [39–42]. In contrast, *Shigella flexneri* and *Yersinia enterocolitica*, where presence of the T3SS mainly gives a benefit to individual cells (invasion of the cytosol of epithelial cells and evasion of the immune response, respectively) [37,43], were so far found to be completely T3SS-positive, both in terms of expression and assembly of injectisomes [42,44–46] and their activation [47].

However, these results were gained under standard laboratory conditions typically focusing on the establishment of an infection at low bacterial densities. It is conceivable that this picture changes under conditions mimicking later stages of infection, where bacteria may benefit from suppressing T3SS assembly and activity, allowing them to replicate extracellularly in their established niches or to disseminate. Increased local bacterial density, a common aspect of later-stage infection environments, might therefore influence the activity of the T3SS.

In this study, we tested this hypothesis and found that T3SS assembly, activation, and bacterial adhesion are governed by cell density. This effect is clearly displayed in *Y. enterocolitica* microcolonies, where only a thin layer of cells at the rim of the colony are T3SS-positive, but can also be replicated and analyzed in liquid cultures. The density-driven downregulation of the T3SS is reversible, highly specific, and distinct from stationary phase adaptation. At higher bacterial densities, we found a > 10 times increased concentration of the regulatory RNA CsrC, which sequesters the RNA-binding protein CsrA. In the closely related *Y. pseudotuberculosis* system, CsrA sequestration was found to decrease the expression of the main T3SS transcription factor VirF [55]. Accordingly, we found that expression of VirF *in trans* overrides the effect. We propose that the highly specific T3SS regulatory phenotype discovered in this study increases the chances of bacterial dissemination during and after infection by creating a T3SS-negative, non-adhesive subpopulation that can replicate and disseminate.

## Results

### Type III secretion activity is confined to the edges of *Y. enterocolitica* microcolonies

To investigate the regulation of T3SS activity under conditions relevant at later stages of infection, we imaged microcolonies of *Y. enterocolitica* Δ*sctW* at host temperature (37°C). T3SS activation can be monitored using the strong upregulation of the effector YopE upon T3SS activation [47–50]. To ensure a high sensitivity and temporal resolution of the reporter, we used the fast-folding fluorescent protein sfGFP with a SsrA degradation tag [51]. The resulting P$_{yopE}$-*sfGFP-ssrA* reporter system allowed for accurate temporal mapping of T3SS activation (S1 Fig). Strikingly, confocal microscopy scanning of the microcolony showed T3SS activation exclusively at the periphery of the colony (Figs 1b, S2), suggesting a strong repression of secretion towards the center.

### Type III assembly and activity are affected by bacterial cell density

To investigate if the spatial regulation of T3SS activation observed in Fig 1b was linked to the different cell densities perceived across the microcolony, we tested the effector secretion of *Y. enterocolitica* liquid cultures incubated under secreting conditions (37°C, low [Ca$^{2+}$]) at increasing cell densities. As *Y. enterocolitica* can reach optical densities at 600 nm (OD$_{600}$) of 15–20, the ODs at the inoculum (OD$_{in}$) used in this experiment correspond to the exponential phase growth

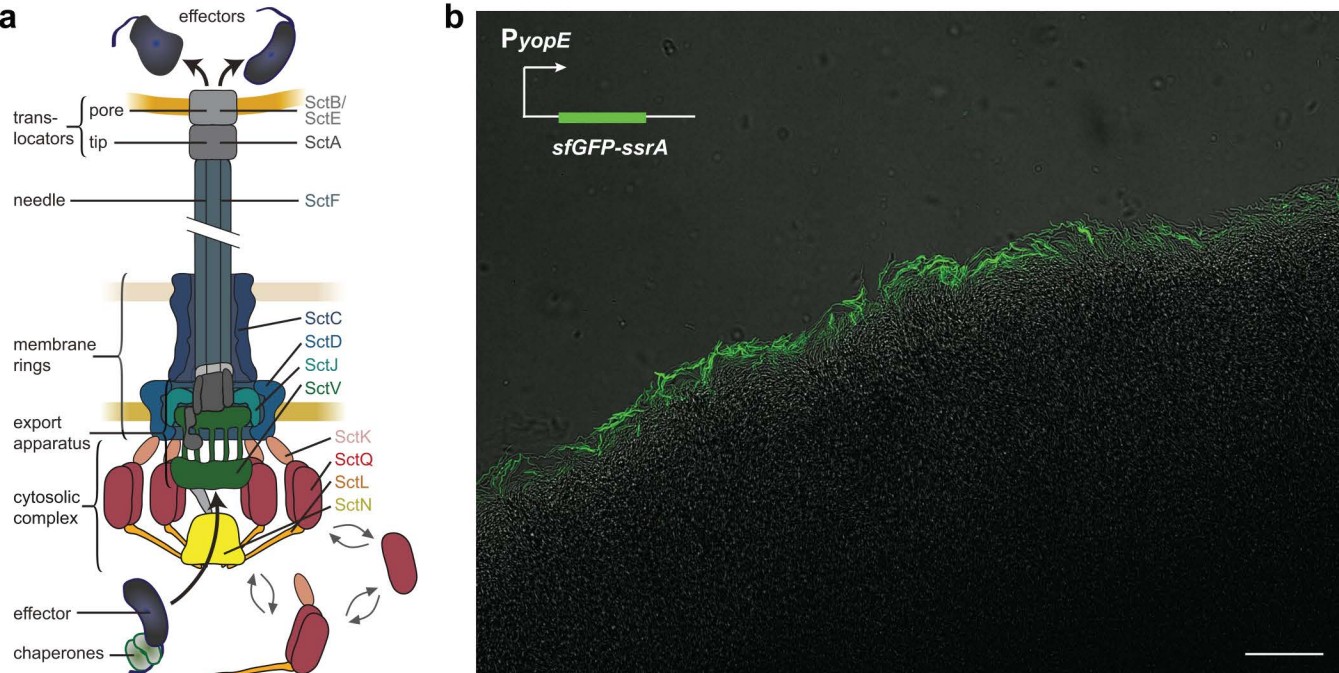

**Fig 1. T3SS activation is only detected at the edge of *Y. enterocolitica* microcolonies.** a) Schematic depiction of the T3SS injectisome (adapted from [52]); right, common nomenclature names of main T3SS components [53,54]. b) Confocal microscopy image of a *Y. enterocolitica* Δ*sctW* P*yopE*-*sfGFP-ssrA* microcolony section at 37°C. T3SS activity, which results in a strong upregulation of the yopE promoter and cellular fluorescence, is only detected at the edge of the microcolony. Scale bar, 50 μm, *n* = 3.

in non-secreting bacteria (S3 Fig). Cultures inoculated to an initial $OD_{600}$ of 0.1 ($OD_{in}$=0.1), a condition regularly used for secretion experiments [42,47,48,55], which we refer to as reference hereafter, secreted large amounts of effector proteins. Increasing densities gradually reduced secretion levels, up to a complete inhibition of T3SS function at higher bacterial densities ($OD_{in}$=1.5) (Fig 2a). This establishes an inverse correlation between T3SS secretion and cell density. To describe this unexpected phenotype in more detail, we next tested T3SS activation over time by quantifying the P*yopE-sfGFP-ssrA* activity. In line with the previous results, the reporter showed a weaker and shorter T3SS activation with increasing bacterial densities, again up to a complete loss of activation at higher densities (Figs 2b, S4). To determine whether already the assembly of the T3SS machinery was impacted by cell density, the number of injectisomes was quantified by fluorescence microscopy at a single-cell level in a strain expressing a functional EGFP-SctQ fusion [44]. The analysis showed a significant reduction in the number of fluorescent foci corresponding to assembled injectisomes at higher densities when compared to the reference (Fig 2c). Like protein secretion itself, T3SS assembly was almost fully repressed at higher cell densities. As expected, a certain degree of heterogeneity was observed under all conditions tested, but neither a bimodal distribution, nor distinct subpopulations were observed for T3SS assembly or activation in single-cell experiments (Fig 2c,d), indicating a general downregulation of the T3SS. The results highlight a strong inverse relationship between cell density and T3SS assembly and activation, which we refer to as density-dependent downregulation of the T3SS (d³T3).

## The density-dependent T3SS inhibition is specific and reversible

To probe if the observed correlation between bacterial density and the T3SS is specific or part of a more global regulatory effect, we compared the total proteome of bacteria with $OD_{in}$=1.5 to the reference ($OD_{in}$=0.1). Remarkably, 24 of the

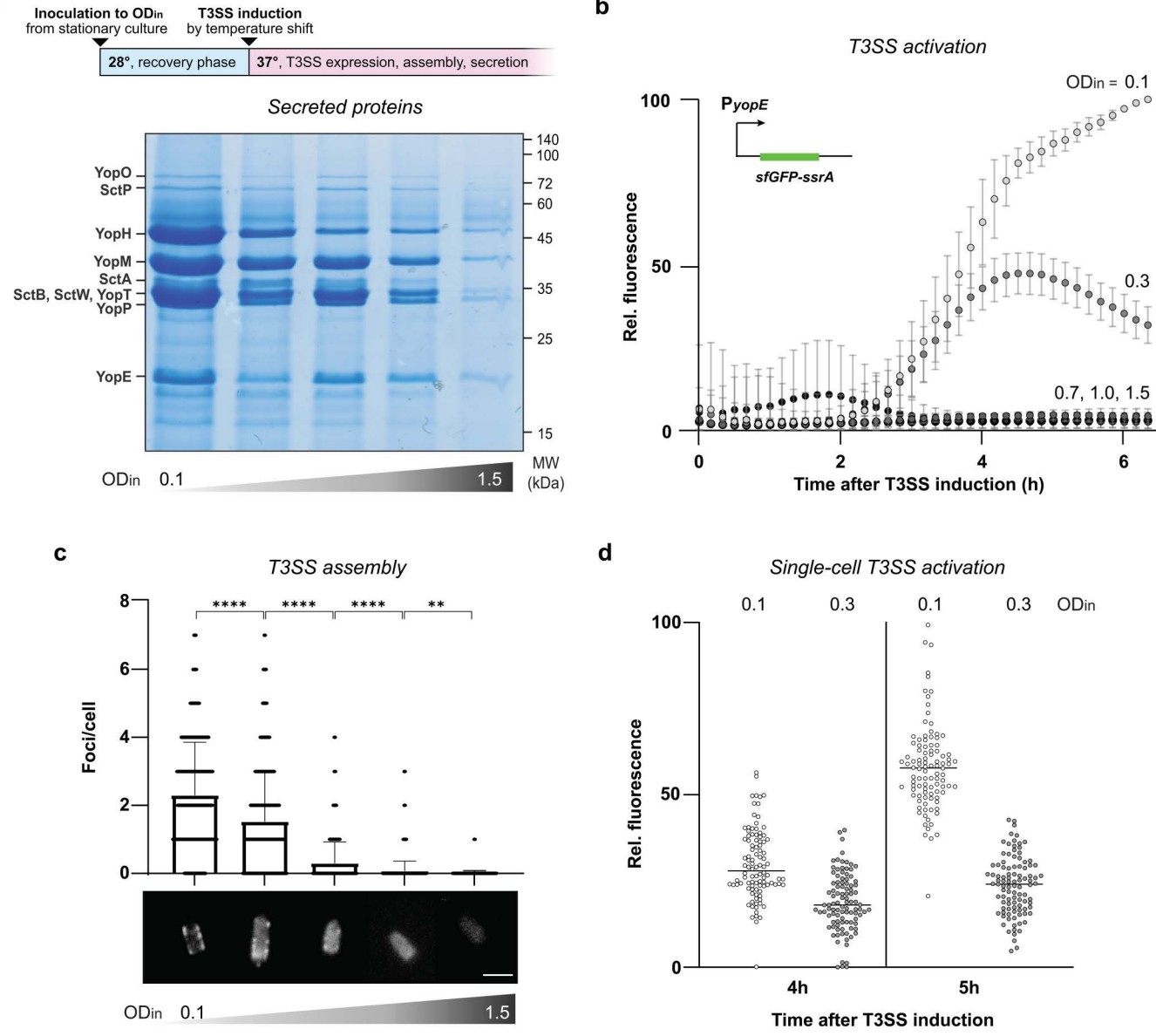

**Fig 2. Assembly and activity of the T3SS are significantly reduced at higher bacterial densities.** a) Top, graphical depiction of secretion assay; bottom, effectors secreted by the T3SS at increasing $OD_{in}$ (left to right: 0.1, 0.3, 0.7, 1.0, 1.5). Left, assignment of effectors; right, molecular weight (MW) standards. b) T3SS reporter assay (PyopE-sfGFP-SsrA) [47,56] at increasing ODin values as in a). Intensities are adjusted for OD600 and displayed relative to the highest OD-adjusted intensity value, growth curves of cultures used in these experiments are displayed in S5 Fig. c) Quantification of EGFP-SctQ foci per bacterium, corresponding to assembled injectisomes, visualized by fluorescence microscopy at increasing $OD_{in}$ as in b). Number of measurements and foci counts listed in S1 Table. Sample micrographs depicted at bottom, scale bar, 2 μm. d) Single-cell T3SS reporter assay (PyopE-sfGFP-SsrA) of Yersinia cells to assess the heterogeneity of T3SS activation at the indicated low and intermediate activation levels (ODin=0.1, 0.3, as indicated). Measurements were taken 4h and 5h post-secretion induction, intensities are displayed as in b). Each dot represents a single cell, black bars represent the average fluorescence intensity. n = 3 for all panels. Gel image in a) shows representative result; for c, n cells = 300; for d, n cells = 100, statistical analysis was performed using a non-parametric t-test, ****, p < 0.0001, whiskers denote standard deviation.

25 most strongly downregulated proteins were directly T3SS-related (marked by red dots in Fig 3a), showing that the effect is highly specific for the T3SS. All T3SS proteins were highly downregulated (3–256-fold, $p < 0.001$ for all detected components), except for the negative regulators YscM1 and YscM2, which were upregulated, and their chaperone SycH, whose expression remained stable (Fig 3a, Table 1). All T3SS gene classes – structural components, regulators, effectors and chaperones – were similarly affected (S6 Fig, Table 1). In contrast, expression of other proteins encoded on the pYV virulence plasmid was largely unaltered (S2 Table). This indicates that the downregulation is not caused by a change in plasmid copy number or overall protein expression from the plasmid. Instead, it suggests a direct effect of bacterial densities on T3SS expression that is distinct from stationary phase responses occurring at higher density. In line with this interpretation, proteins involved in stationary phase response, which are highly expressed at stationary phase, were not significantly upregulated at $OD_{in} = 1.5$ (S3 Table). Taken together, our data unanimously point towards a specific T3SS repression mechanism driven by cell density rather than a broader cellular reorganization.

We then tested if the d³T3 effect is reversible. Bacterial cultures were inoculated at reference or higher OD ($OD_{in} = 0.1$ or 1.5) and then concentrated or diluted to the OD of the respective other culture prior to induction of T3SS expression by a temperature shift to 37°C in the absence of $Ca^{2+}$. Bacteria were resuspended in their respective centrifugation supernatant to avoid possible effects of fresh medium on the analysis. Activation of the T3SS was measured as P*yopE-sfGFP-ssrA* reporter activity for single bacteria. Increasing the density of initially dilute cultures strongly suppressed T3SS activation. Likewise, diluting the initially more concentrated cultures almost completely restored T3SS activation (Fig 3b). We also tested a possible role of the medium composition (e.g., by limitation of nutrients or presence of extracellular negative regulators at higher densities). Secretion at low bacterial density was consistently slightly reduced (by 30–40%) by the addition of spent medium from high-density cultures, indicating a possible effect of nutrient limitation or changed pH. In contrast, secretion was not activated at high densities by provision of fresh medium (Fig 3c). Together, these results highlight a direct and reversible influence of bacterial density on T3SS secretion.

**Global stationary phase regulators have a minor effect on T3SS assembly and function**

We next aimed to determine the cause for the previously unknown regulatory phenotype described above. To investigate a possible role of known density-dependent global regulators of bacterial physiology and behavior in the d³T3, we studied the influence of quorum sensing (QS), the stationary phase/stress response alternative sigma factor RpoS, and the stringent response regulators RelA and SpoT. In *Yersinia*, both homologs of LuxI/R (*yenI/R*) and LuxS have been characterized, although no clear evidence of functional LuxS-dependent AI-2 signaling has been provided so far. While QS in *Yersinia* influences, among others, motility and biofilm formation [56–58], its role in bacterial virulence is less clear. We performed a secretion assay in a *Y. enterocolitica* QS double mutant lacking the genes for YenI and the AI-2 kinase LsrK. The secretion pattern of the mutant showed no significant impact of QS on secretion under the conditions tested (Figs 4a, S7). Next, we tested the influence of the stationary phase/stress response RNA polymerase sigma factor RpoS on d³T3. RpoS is known to positively regulate motility and biofilm formation in many Gram-negative bacteria [59,60], but its impact on *Yersinia* virulence is unclear. The activity of the T3SS at $OD_{in}$ 0.1 and 1.5 was not influenced by the deletion or overexpression of RpoS (Fig 4b). Finally, we tested the role of RelA [61,62] and SpoT [63,64], activators of the stringent response, which is a key process in stationary phase adaptation. Similar to the previous experiments, deletion of RelA or SpoT did not restore secretion at higher bacterial densities (S8 Fig). These results further support the hypothesis that the d³T3 is a specific phenotype rather than a part or consequence of stationary phase adaptation.

**The transcriptional regulator VirF is the main target of the density-dependent T3SS regulation**

Based on the specificity of the d³T3 (Fig 3a), the uniform influence on all T3SS components (Table 1, S6 Fig), and the limited role of the tested global regulators (Figs 4, S8), we hypothesized the target of the repression to be a general regulator of the T3SS. Expression and assembly of the injectisome are controlled mainly by the AraC-like transcriptional

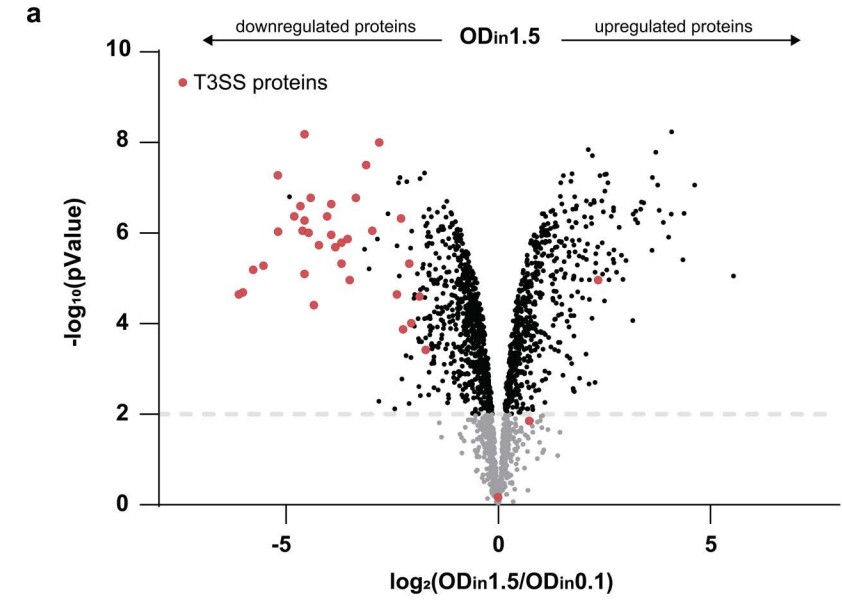

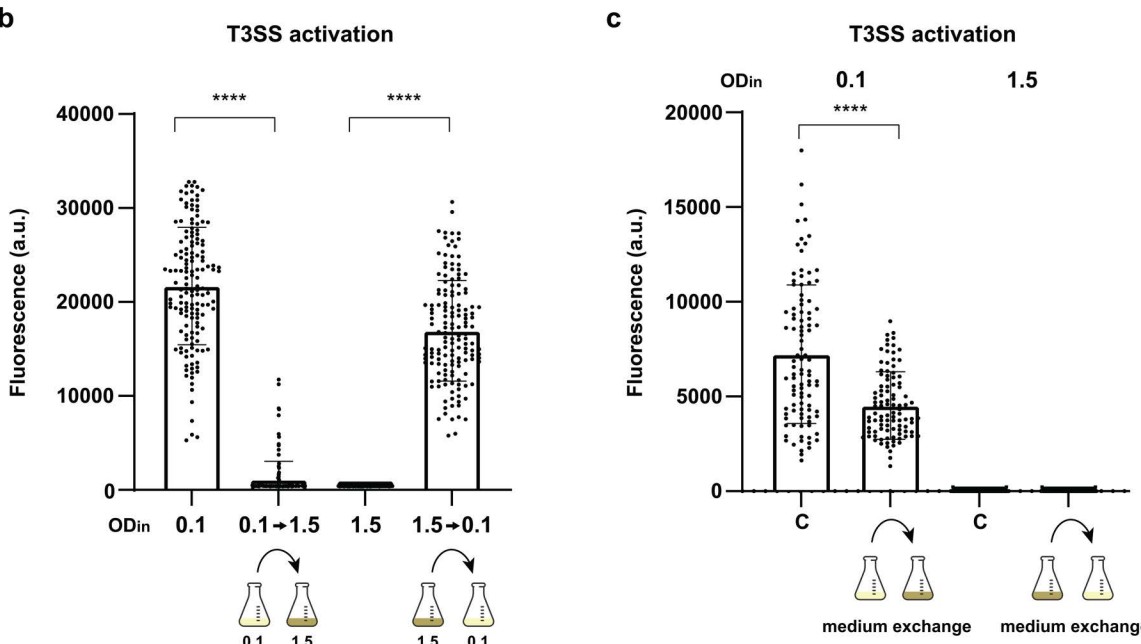

**Fig 3. The density-dependent downregulation of the T3SS is specific and reversible.** a) Volcano plot showing differences in the level of *Y. entero-colitica* proteins between cultures grown under secreting conditions at $OD_{in}$ 1.5 and 0.1. All proteins with at least three detected peptides under both conditions are displayed; T3SS-related proteins are marked by larger red dots. b) T3SS reporter assay (PyopE-sfGFP-ssrA) in strains with ODin=0.1 or 1.5 and concentration or dilution at the time of the temperature shift to 37°C, as indicated. Measurements were performed 150 min after dilution/concentration. c) T3SS reporter assay (PyopE-sfGFP-ssrA) in strains with ODin values as indicated. Medium exchange at the time of the temperature shift to 37°C as depicted; C indicates constant incubation without medium change. Measurements were performed 150 min after medium exchange. n = 3 biological replicates for all panels. In panels b) and c), n cells = 150, each spot represents a single measurement, whiskers denote standard deviation. Statistical analysis was performed using a non-parametric t-test; ****, $p < 0.0001$.

**Table 1. Density-dependent regulation of expression of T3SS components.**

| Protein | Log$_2$ intensity ratio | p value | Individual log$_2$ intensity values | | | | | | # pept. |
|---|---|---|---|---|---|---|---|---|---|
| | | | OD$_{in}$ = 0.1 | | | OD$_{in}$ = 1.5 | | | |
| Rod/ washer SctI | -8.00 | 6.9E-08 | 27.30 | 27.79 | 27.78 | 19.42 | 19.56 | 19.89 | 10 |
| Unknown T3SS-associated (ORF91A) | -7.64 | 1.6E-07 | 28.80 | 28.80 | 28.90 | 21.05 | 20.71 | 21.54 | 10 |
| IM ring component SctD | -6.29 | 2.4E-05 | 29.83 | 29.83 | 29.89 | 22.59 | 23.72 | 24.38 | 20 |
| Exp.app. component SctV | -6.24 | 2.3E-05 | 31.26 | 31.23 | 31.34 | 24.29 | 24.81 | 26.01 | 76 |
| Exp.app. component SctR | -5.93 | 4.6E-07 | 25.81 | 25.83 | 25.98 | n.d. | n.d. | n.d. | 5 |
| Effector YopO | -5.84 | 6.4E-06 | 31.16 | 31.23 | 31.24 | 24.63 | 25.83 | 25.65 | 25 |
| Chaperone SycO | -5.64 | 9.1E-07 | 30.23 | 30.27 | 30.26 | 24.65 | 24.99 | 25.44 | 14 |
| Translocator SctE (YopB) | -5.53 | 5.3E-06 | 33.23 | 33.97 | 33.42 | 27.54 | 28.04 | 28.44 | 39 |
| Needle subunit SctF | -5.20 | 5.4E-08 | 30.92 | 30.91 | 30.93 | 25.68 | 25.52 | 25.96 | 13 |
| Translocator SctA (LcrV) | -4.80 | 4.1E-07 | 33.40 | 33.35 | 33.39 | 28.29 | 28.56 | 28.90 | 58 |
| Secretin/ OM ring SctC | -4.66 | 2.6E-07 | 30.90 | 30.97 | 30.98 | 26.02 | 26.30 | 26.56 | 38 |
| Regulator YscX | -4.59 | 5.4E-07 | 27.99 | 28.17 | 28.01 | 23.18 | 23.45 | 23.77 | 12 |
| IM ring component SctJ | -4.59 | 9.2E-07 | 31.26 | 31.25 | 31.32 | 26.46 | 26.52 | 27.09 | 35 |
| Chaperone YscE | -4.56 | 6.8E-09 | 29.74 | 29.73 | 29.92 | 25.25 | 25.17 | 25.29 | 13 |
| Translocator SctB (YopD) | -4.54 | 7.4E-06 | 32.14 | 32.94 | 32.27 | 27.69 | 27.81 | 28.23 | 71 |
| Chaperone YscB | -4.52 | 9.5E-07 | 30.65 | 30.60 | 30.78 | 25.83 | 26.15 | 26.49 | 13 |
| Effector YopT | -4.48 | 7.4E-03 | 23.29 | 23.33 | 23.28 | n.d. | 16.56 | n.d. | 5 |
| Chaperone YscG | -4.46 | 1.7E-07 | 29.24 | 29.19 | 29.29 | 24.59 | 24.70 | 25.04 | 8 |
| Regulator LcrG | -4.25 | 1.9E-06 | 30.42 | 30.40 | 30.47 | 25.80 | 26.20 | 26.53 | 28 |
| Chaperone SycT | -4.25 | 4.0E-07 | 24.33 | 23.33 | 23.41 | n.d. | n.d. | n.d. | 3 |
| Cytosolic component SctK | -4.08 | 1.2E-06 | 28.67 | 28.68 | 28.75 | 24.39 | 24.58 | 24.90 | 18 |
| Lipoprotein YlpA | -3.98 | 1.6E-06 | 30.97 | 30.94 | 30.90 | 26.89 | 27.28 | 27.50 | 21 |
| Exp.app. component SctU | -3.94 | 2.4E-07 | 26.93 | 27.02 | 27.09 | 22.81 | 23.02 | 23.40 | 16 |
| Cytosolic component SctL | -3.94 | 8.1E-07 | 29.00 | 29.06 | 29.12 | 24.90 | 25.13 | 25.33 | 16 |
| Regulator YscY | -3.92 | 2.1E-06 | 26.31 | 26.29 | 26.40 | 22.55 | 22.09 | 22.60 | 5 |
| Chaperone SycD | -3.87 | 1.6E-06 | 32.33 | 32.35 | 32.34 | 28.12 | 28.48 | 28.80 | 22 |
| Transcriptional activator VirF (LcrF) | -3.70 | 4.7E-06 | 28.76 | 28.79 | 28.85 | 24.76 | 25.03 | 25.51 | 15 |
| Cytosolic ATPase SctN | -3.56 | 1.1E-05 | 30.93 | 30.93 | 31.02 | 26.92 | 27.53 | 27.76 | 46 |
| Effector YopQ | -3.54 | 1.3E-06 | 29.25 | 29.44 | 29.11 | 25.51 | 25.69 | 25.97 | 17 |
| Chaperone SycN | -3.34 | 1.8E-07 | 29.71 | 29.65 | 29.97 | 26.39 | 26.42 | 26.51 | 9 |
| Unknown T3SS-associated (ORF181) | -3.27 | 1.0E-08 | 25.65 | 25.60 | 25.70 | 22.87 | 22.86 | 22.80 | 8 |
| Gatekeeper TyeA | -3.11 | 3.4E-08 | 29.16 | 29.16 | 29.20 | 25.99 | 26.02 | 26.19 | 9 |
| Secreted protein YscH/YopR | -2.98 | 9.7E-07 | 27.09 | 27.10 | 26.92 | 24.00 | 23.88 | 24.27 | 5 |
| Effector YopE | -2.47 | 5.2E-04 | 24.95 | 24.83 | 25.09 | 22.73 | 21.76 | 22.96 | 6 |
| Needle length regulator/ ruler SctP | -2.40 | 2.3E-05 | 30.30 | 30.37 | 30.36 | 27.59 | 27.97 | 28.26 | 45 |
| Cytosolic component SctO | -2.30 | 1.4E-04 | 25.40 | 25.39 | 25.39 | 22.63 | 23.10 | 23.55 | 9 |
| Regulator LcrR | -2.27 | 5.2E-07 | 27.41 | 27.43 | 27.55 | 25.11 | 25.14 | 25.33 | 6 |
| Chaperone SycE | -2.10 | 4.5E-06 | 30.09 | 30.15 | 30.11 | 27.84 | 27.96 | 28.25 | 8 |
| Exp.app. component SctT | -2.09 | 1.5E-02 | 22.65 | 22.34 | 22.24 | n.d. | n.d. | n.d. | 4 |
| Cytosolic component SctQ | -2.06 | 1.1E-04 | 29.58 | 29.58 | 29.63 | 27.10 | 27.71 | 27.82 | 13 |
| Pilotin SctG (YscW) | -1.90 | 2.4E-05 | 28.46 | 28.44 | 28.39 | 26.35 | 26.41 | 26.83 | 9 |
| Secreted gatekeeper SctW (YopN) | -1.73 | 3.8E-04 | 30.68 | 31.00 | 30.70 | 28.66 | 29.10 | 29.42 | 25 |
| Chaperone SycH | -0.02 | 7.4E-01 | 29.65 | 29.63 | 29.66 | 29.57 | 29.55 | 29.75 | 21 |
| Secreted neg. regulator YscM1 | 0.81 | 1.4E-02 | 27.53 | 27.52 | 26.86 | 28.34 | 28.06 | 27.93 | 11 |
| Secreted neg. regulator YscM2 | 2.36 | 1.1E-05 | 26.01 | 26.07 | 25.96 | 28.64 | 28.40 | 28.09 | 9 |

Label-free quantitative mass spectrometry of T3SS components in the total proteome of a ΔHOPEMTasd wild-type strain at the different growth conditions indicated. Proteins are sorted by the ratio of intensity at OD$_{in}$ 1.5/OD$_{in}$ 0.1 (red shades: downregulated at OD$_{in}$ 1.5, blue shades: upregulated at OD$_{in}$ 1.5), which the p values refer to as well. Exp.app., IM export apparatus; # pept., number of total detected peptides. Note that for some effectors, the mutations in the ΔHOPEMTasd strain allow the detection of a subset of peptides (see S6 Table for details). n.d., not detected; imputed values were used for the statistical analysis in these cases.

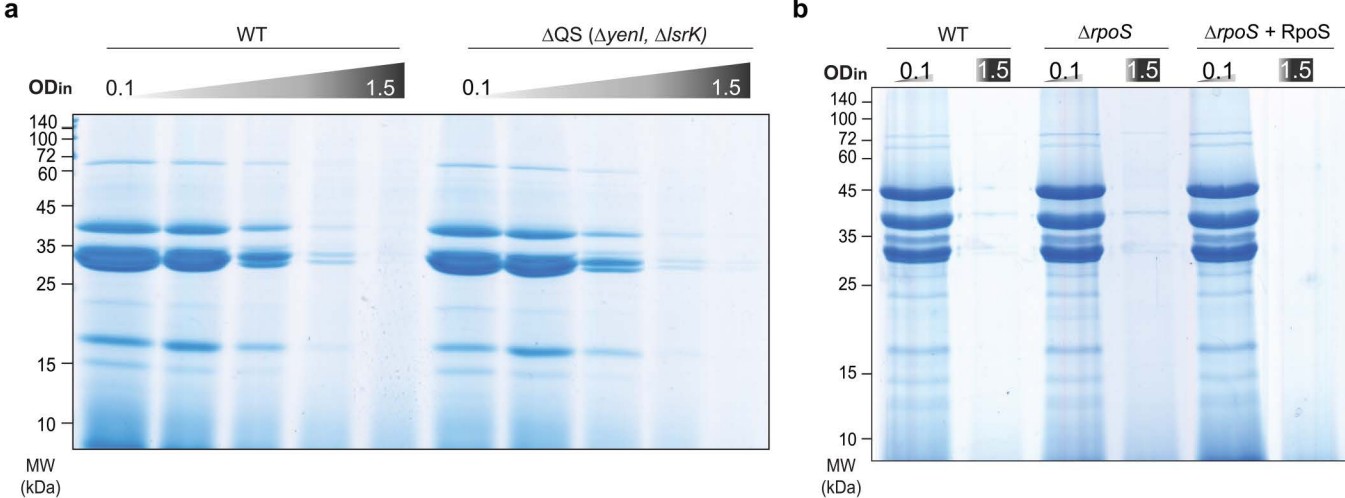

**Fig 4. Quorum sensing and the alternative sigma factor RpoS do not significantly contribute to the density-dependent downregulation of the T3SS.** a) Secretion assay of wild-type (WT) and quorum sensing mutant strains at different $OD_{in}$ as used in [Fig 1](left to right, $OD_{in}$=0.1, 0.3, 0.7, 1.0, 1.5). b) Secretion assay of the indicated strains at $OD_{in}$ 0.1 and 1.5, as displayed. For *in trans* complementation of the *rpoS* mutant from pBAD::RpoS, RpoS expression was induced by the addition of 0.2% arabinose at the time of the shift to 37°C. $n = 3$, gel images show representative results.

activator VirF (S9a Fig). In the absence of VirF, the *Yersinia* T3SS is inactive and its components are strongly repressed [19,65,66] (S9b Fig). VirF was among the proteins that were significantly downregulated at high cell densities (-12.64-fold, $p < 10^{-5}$, Table 1, S6 Fig). To test if this downregulation is the causal factor for the d³T3, we expressed VirF from a plasmid at $OD_{in}$=1.5. Additional expression of VirF was indeed sufficient to restore T3SS activity (Fig 5a), indicating that T3SS repression at higher cell densities largely is a consequence of reduced VirF activity. Based on these results, added VirF expression should lead to uniform T3SS activity in microcolonies. To test this, we imaged a microcolony grown as previously (Fig 1b), but formed by *Y. enterocolitica* expressing additional VirF from plasmid. In line with this hypothesis, the resulting increase in VirF levels abolished the downregulation of T3SS activity in the center of the colony, leading to a homogenously T3SS-active microcolony (Figs 5b, S10).

**Increased concentrations of the post-transcriptional regulators CsrB and CsrC at higher bacterial density**

We next investigated how cell density influences VirF levels. VirF is encoded in the *sctG-virF* operon (Fig 6a). *sctG* encodes the pilotin protein, which assists the assembly of the T3SS secretin ring in the OM [67,68], but also has additional post-assembly functions [69]. Interestingly, it is the only T3SS component whose expression is largely VirF-independent [18,19,70]. Total proteome analysis and immunoblots showed that, while SctG is affected by higher cell densities, the effect is significantly weaker than for VirF or other T3SS components (Figs 6b, S11, Table 1). Fluorescence microscopy confirmed this on a single-cell basis (Fig 6c). These results suggest that the regulatory region directly upstream of *virF*, rather than the promoter of the *sctG-virF* operon, is the target of d³T3. To discriminate between regulation of transcription of *sctG-virF* and post-transcriptional regulation specifically affecting *virF*, we measured the transcript abundance of *sctG* and *virF* by quantitative reverse transcription polymerase chain reaction (RT-qPCR). Mirroring the expression data, *virF* transcript levels were more strongly reduced at higher densities than *sctG*, pointing towards a cell density-driven post-transcriptional targeting of the *virF* mRNA (Fig 6d). A known regulator of *virF* transcript abundance is the CsrABC

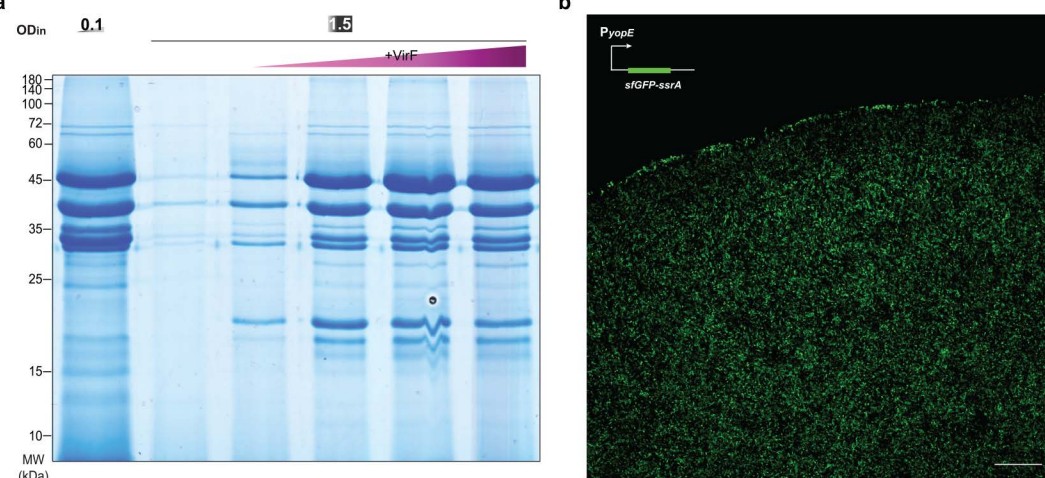

**Fig 5. Additional expression of the transcriptional regulator VirF at higher densities restores T3SS secretion.** a) Secretion assay showing effector secretion at ODin=0.1 (left) and 1.5 (right). Lanes 3-6, increasing expression of VirF in trans (arabinose concentrations of 0.001, 0.003, 0.008, 0.03%, respectively). b) Confocal microscopy image of a Y. enterocolitica ΔsctW PyopE-sfGFP-ssrA microcolony additionally expressing VirF (induced by 0.2% arabinose) at 37°C. T3SS activity, visualized by the sfGFP signal, is detected throughout the colony. Scale bar, 50 μm; n = 3 for all panels, images display representative results.

pathway. In *Y. pseudotuberculosis*, CsrA modulates T3SS activation via *virF* transcript regulation [55]. CsrA binds to the *sctG-virF* mRNA intergenic region as a dimer, stabilizing the *virF* transcript by reducing its turnover rate [55,71] (Fig 6e). The two conserved regulatory non-coding RNAs CsrB and CsrC (S12 Fig) sequester CsrA, preventing it from binding and stabilizing the *virF* transcript. The *virF* upstream and coding region are 98% identical between *Y. enterocolitica* and *Y. pseudotuberculosis* (S13 Fig). Expression levels of the three components of the regulatory system are linked via feedback loop mechanisms [72] (S14 Fig), complicating the investigation by genetic manipulations of the system. To investigate a possible involvement of the CsrABC system in the d³T3, we therefore measured CsrB and CsrC levels at different densities and observed a significant upregulation of CsrB and especially CsrC at higher bacterial densities (Fig 6f). As CsrA levels remain relatively constant at the different bacterial densities (S4 Table), this suggests a subsequent increased sequestration of CsrA by CsrC, which may underlie the *virF* downregulation that causes d³T3 at higher bacterial densities.

## Host-cell adhesion is affected by cell density

Adhesion to host cells is a key virulence factor for many bacterial pathogens and a prerequisite for the utilization of the T3SS. Accordingly, the trimeric *Yersinia* adhesin YadA, which is co-encoded with the T3SS on the pYV virulence plasmid, is also regulated by VirF [75,76]. In line with this co-regulation, total proteome analysis showed a strong downregulation of YadA at higher cell densities ($OD_{in}$ 1.5) (31.3-fold, $p < 10^{-6}$) (Fig 7a) – in fact, YadA is the only non-T3SS protein within the 25 proteins most affected by the d³T3 (Figs 3a, S6). To test whether this downregulation leads to a decreased adhesion to eukaryotic cells, which might promote the detachment and dissemination of bacteria at higher cell densities occurring at the later stages of infection, we performed an attachment assay to adherent HeLa cells. Indeed, bacteria incubated at higher densities prior to the assay attached significantly less than the reference ($OD_{in}$=0.1); the reduction almost mimicked the behavior of bacteria lacking *yadA*, which were used as a control (Fig 7b). These results indicate a coordinated downregulation of T3SS activity and bacterial adhesion at higher densities.

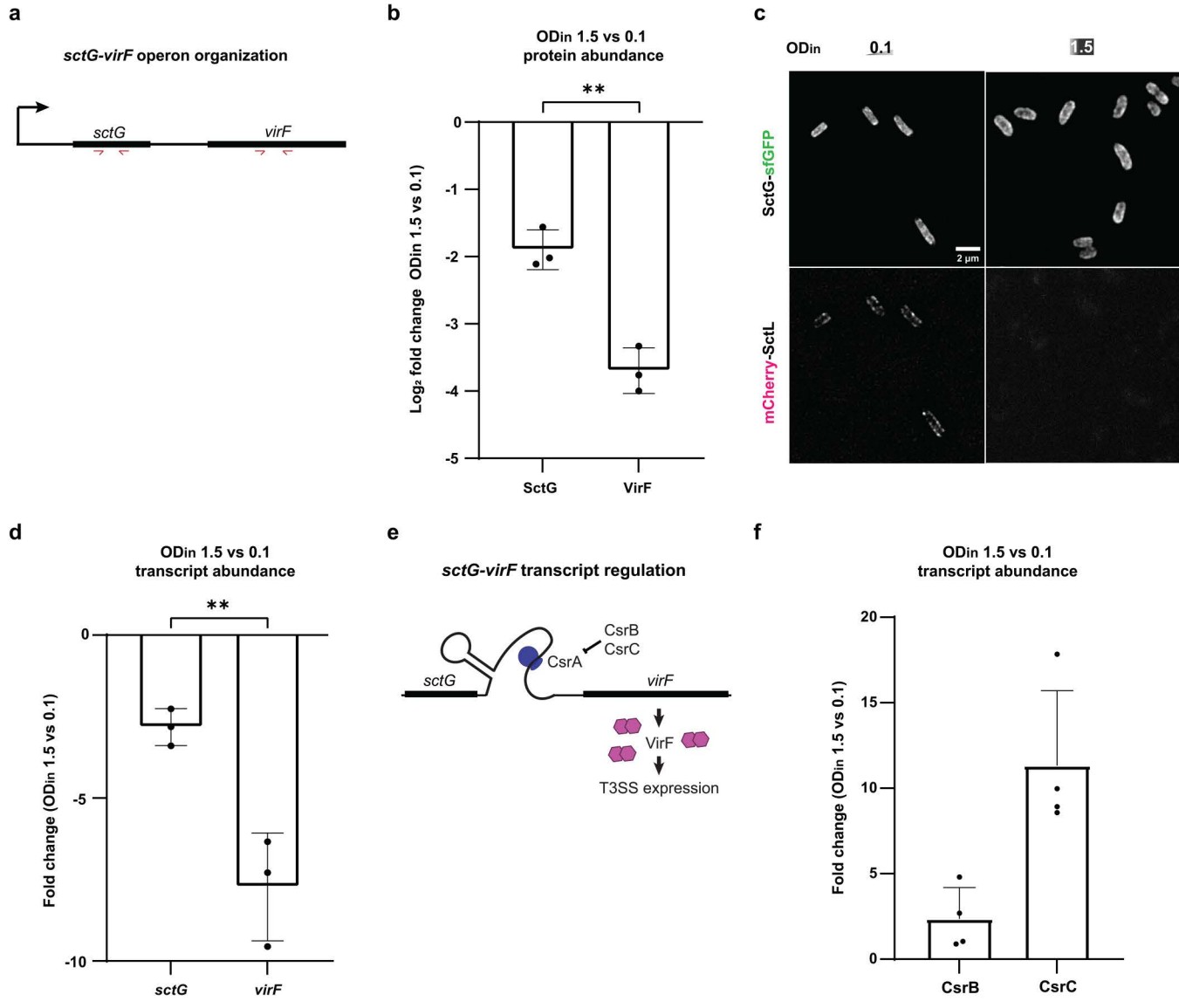

**Fig 6. Increased abundance of the regulatory RNAs CsrBC at high bacterial density and posttranscriptional control of the virF transcript.** a) Organization of the *sctG-virF* operon, bold lines depict coding regions [18,69,73], red arrows indicate the position of RT-PCR primers. b) Protein levels of SctG and VirF at $OD_{in}$ 1.5 vs. 0.1 (displayed as log2 fold change), as measured by quantitative proteomics. c) Fluorescence microscopy of *SctG-sfGFP* (top) and the VirF-dependent T3SS component mCherry-SctL, both expressed from their native genetic background, at $OD_{in}$ = 0.1 (left) or 1.5 (right). n = 3, micrographs display representative results. d) Transcript levels of sctG and virF at $OD_{in}$ 1.5 vs. 0.1, as measured by reverse transcriptase quantitative PCR (see panel A). e) Schematic depiction of the influence of the CsrABC system on VirF production [26,55,74]. The regulatory RNAs CsrB and CsrC sequester CsrA, which in the closely related *Y. pseudotuberculosis* system (S13 Fig) prevents it from binding and stabilizing the *virF* transcript. f) Transcript levels of CsrB and CsrC at $OD_{in}$ 1.5 vs. 0.1, as measured by reverse transcriptase quantitative PCR. In b), d) and f), points indicate individual biological replicates, whiskers denote standard deviation. Statistical analysis was performed using a non-parametric t-test, **, $p < 0.01$.

## Discussion

To establish an infection, bacterial pathogens initially need to enter the host organism and defend themselves against the host immune system on the way to their replication niches. However, after this initiation of infection, it is equally important for the bacteria to propagate in their niches and disseminate to other areas of the host organism or other hosts, steps that

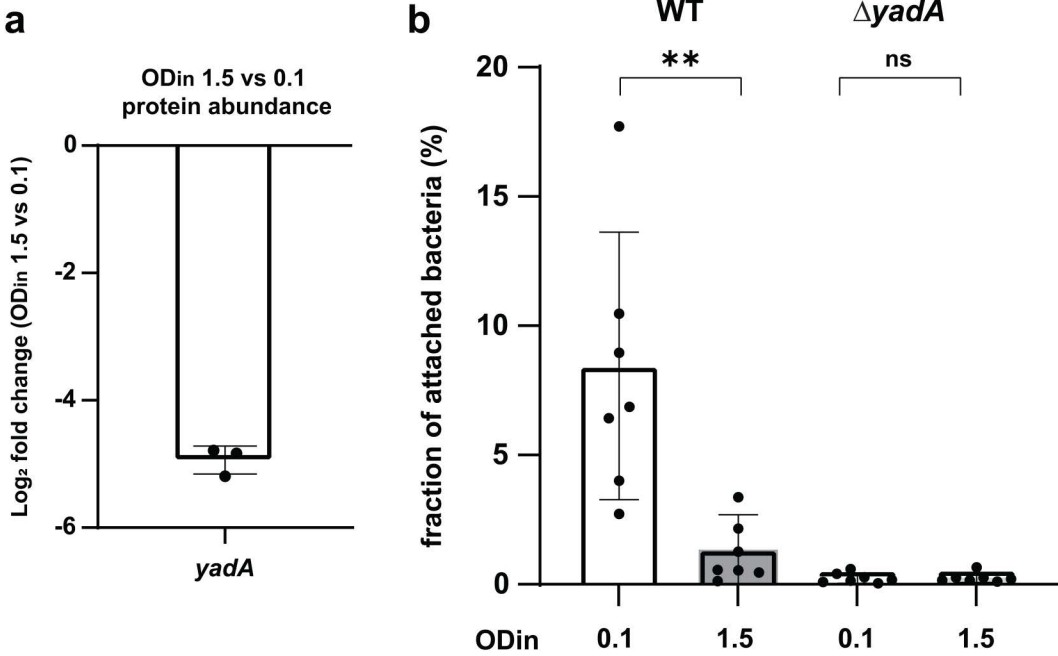

**Fig 7. Host-cell adhesion is significantly downregulated at higher cell densities.** a) Protein levels of YadA at $OD_{in}$ 1.5 vs. 0.1, as measured by quantitative proteomics. Points indicate individual biological replicates, whiskers denote standard deviation. $n = 3$. b) Attachment of wild-type (WT) or Δ*yadA* bacteria to HeLa cells at the conditions indicated ($OD_{in}$=0.1, 1.5) (displayed as fraction of attached bacteria). Points indicate individual biological replicates, whiskers denote standard deviation. Statistical analysis was performed using a non-parametric t-test **, $p < 0.01$, ns, $p > 0.05$.

are often less appreciated and understood. The T3SS is a major virulence factor for important bacterial pathogens including *Salmonella*, *Shigella*, *Yersinia* and pathogenic *E. coli*. Its role in the establishment of infections is well-characterized and includes bacterial uptake and the establishment of inflammation in *Salmonella* [77], inhibition of phagocytosis and inflammation in *Yersinia* and *Shigella* [78,79], and actin pedestal formation and colonization in pathogenic *E. coli* [80]. However, presence of the T3SS may become an obstacle in the later steps of infection, because T3SS activity severely restricts bacterial growth and division across main T3SS model organisms, including *Yersinia* [26,27], *Salmonella* [32], and *Shigella* [34,81]. In this study, we set out to address this important conundrum by investigating the formation and activity of the *Y. enterocolitica* T3SS during microcolony formation. We found that *Yersinia* selectively activates T3SS expression and assembly only in cells at the rim of the microcolony. The phenotype can be explained by an active downregulation of the system at higher local densities, which we characterized in liquid cultures at controlled growth conditions. Our findings contrast with the previous picture of a completely T3SS-positive *Yersinia* population obtained from studies that focused on the initiation of an infection and therefore used lower cell densities [42,45,47,48,55,77]. Proteome analysis showed that the density-dependent T3SS downregulation is a highly specific phenotype that affects all T3SS components and the coregulated adhesion factor YadA, but very few non-T3SS-associated proteins (Figs 3a,S6,Tables 1,S2).

Density-dependent downregulation of the T3SS may be mediated by a pathway involving CsrABC and VirF, the main T3SS transcription factor in *Yersinia*. At higher bacterial densities, higher concentrations of the regulatory RNAs CsrB and CsrC sequester the regulatory protein CsrA. In *Y. pseudotuberculosis*, CsrA was shown to bind to and most likely stabilize the *virF* mRNA, and sequestration of CsrA leads to a gradual inhibition of the expression of VirF [55,71] (Fig 8b). This reduction of VirF levels and, subsequently, all T3SS components at high *Y. enterocolitica* densities (Fig 3a,S6,Table 1), prevents the formation of injectisomes and protein secretion by the T3SS. The post-transcriptional nature of the effect

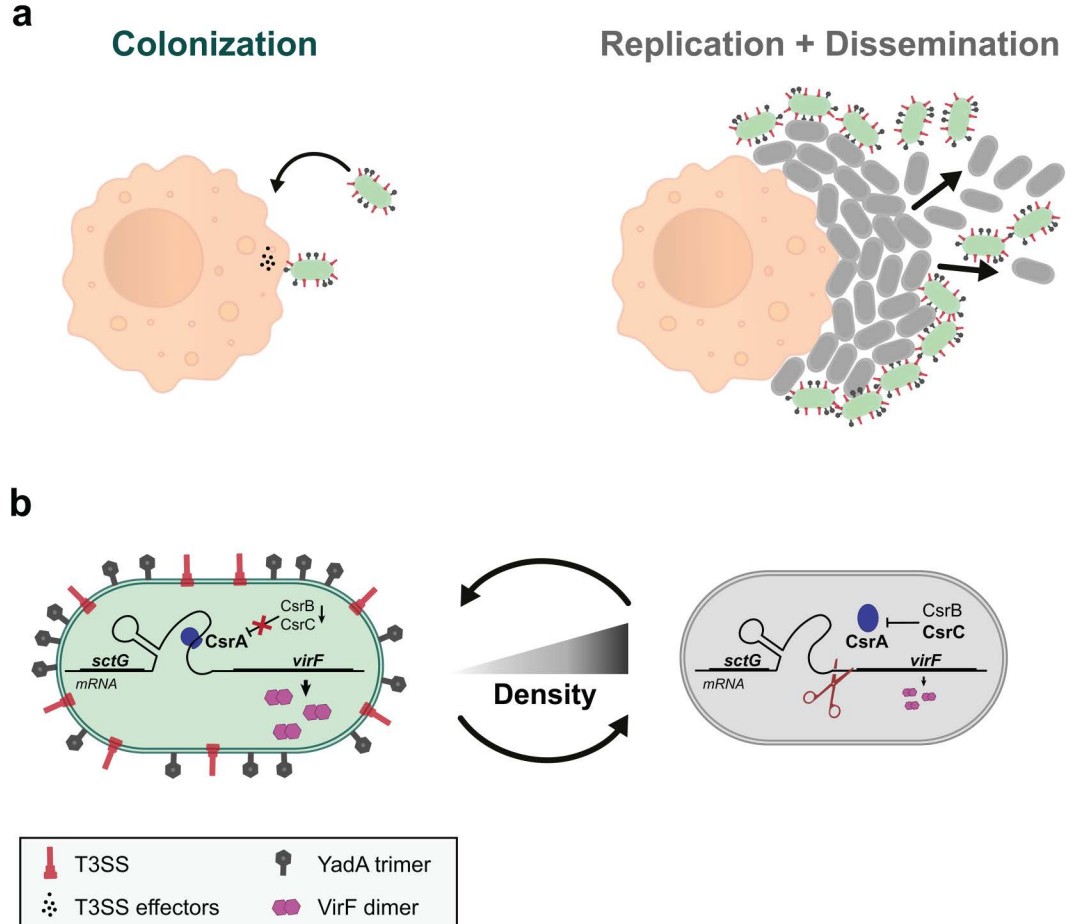

**Fig 8. Model of the influence of local higher cell density on T3SS secretion and on host-cell attachment.** a) At lower cell densities (left), *Yersinia* relies on the adhesion protein YadA (depicted in grey on the bacterial surface) and effector secretion by the T3SS (depicted in red on the bacterial surface) to colonize the host by fighting the host immune system. In a microcolony, where higher cell densities are perceived (right), *Yersinia* specifically downregulates T3SS secretion and YadA adhesion to efficiently replicate and ultimately spread and disseminate. b) Both adhesion and secretion are regulated by the transcription factor VirF. Upon host entry, sensed as 37°C shift, the *virF* transcript is stabilized thanks to the global mRNA regulator CsrA. The efficient binding of CsrA to *virF* mRNA and resulting expression of T3SS components are ensured by the low basal levels of the two CsrA inhibitory RNAs CsrBC [26,55,74]. At higher local cell densities, increased CsrC levels sequester CsrA, which causes a higher turnover of the *virF* transcript, leading to a decreased expression of T3SS components and YadA. The resulting decrease in adhesion and T3SS activity promotes *Yersinia* replication and dissemination at later stages of infection.

on VirF [78] is illustrated by the finding that while *sctG* transcript levels are reduced at higher bacterial densities, this is most likely a secondary effect of *virF* transcript destabilization, as the effect on *virF* levels is significantly stronger (Fig 6b). This regulatory pathway is a main factor in the density-dependent downregulation of the T3SS, which is highlighted by the finding that additional expression of VirF *in trans* restores T3SS activity at higher bacterial densities and leads to homogenously T3SS-active microcolonies.

The CsrABC (carbon storage regulator) system, originally identified as a glycogen synthesis regulator in *E. coli* [79] (S14 Fig), acts as a general regulator of bacterial virulence, motility, quorum sensing and biofilm formation via target mRNA transcript binding in many bacterial species [80–82]. Studies in *Y. pseudotuberculosis* showed that the CsrABC system, which is regulated by complex feedback loops [72], strongly affects expression of the T3SS [26,55,74]. While overall, we found that gene regulation is comparable in our conditions (37°, T3SS-inducing conditions), the effect on

the T3SS is stronger than on other genes affected by CsrABC (Fig 3a, S5 Table). This reflects earlier observations that global regulatory pathways have a less pronounced effect when the T3SS is active, and highlights the specificity of the density-dependent downregulation of the T3SS. Similar to previous studies of CsrABC-based regulation of bacterial virulence, the precise upstream factors that convey the local bacterial density to the regulatory RNAs remain unclear. We experimentally tested the involvement of several candidates, namely stringent response, quorum sensing (QS), the alternative sigma factor RpoS, and factors secreted into or depleted from the medium. There are several known links between QS and the T3SS, such as the inhibitory role of cell-cell communication on *Vibrio harveyi* and *P. aeruginosa* T3SS at high cell density [83,84]. Furthermore, Atkinson and colleagues provided evidence for a negative influence of QS on T3SS secretion in *Y. pseudotuberculosis* during biofilm formation in *Caenorhabditis elegans* [85]. However, we found that a *Y. enterocolitica* strain lacking functionality of both known QS systems still retained the $d^3T3$ (Fig 4a). This lack of influence of the QS on $d^3T3$ is in line with previous studies showing no effect of QS on *Y. enterocolitica* T3SS regulation [56,58]. Similarly, we neither observed a significant downregulation of T3SS activity in spent medium previously incubated at higher densities, nor a recovery of activity at high densities upon provision of fresh medium (Fig 3b,c), showing that any potential factor in the medium that might control T3SS activity would be short-lived. Ultimately, the molecular mechanisms that leads to the upregulation of CsrBC and its direct effect on VirF in *Y. enterocolitica* remain to be determined.

The T3SS is an essential virulence factor for many of the pathogenic bacteria that employ it, such as *Salmonella*, *Shigella* and *Yersinia*. However, its presence and activity are a double-edged sword: While the T3SS allows to manipulate host cells and in this way to evade the immune response (reviewed in [86]), or to ensure the bacterial entry into target cells (reviewed in [87]), its activity severely restricts the bacterial ability to multiply. Accordingly, assembly and activity of the T3SS are tightly regulated. A striking example is the regulated expression of the SPI-1 T3SS in *Salmonella enterica*, where the inflammation caused by the activity of the T3SS in a fraction of bacteria leads to a clearing of bacterial competitors, allowing the T3SS-negative siblings to fill the resulting niche [34,88,89]. A similar effect was previously not observed for *Yersinia*, which is uniformly T3SS-positive at the low densities used for most laboratory experiments [42,47,48,55]. This could be rationalized by the fact that, like in *Shigella*, the T3SS mainly provides an individual advantage by phagocytosis and inflammation inhibition [77,90,91]. However, like other bacteria, *Yersinia* needs to propagate in its niche. We hypothesize that the $d^3T3$, described in this study, allows efficient bacterial replication in colonized niches and the subsequent spreading from these niches and microcolonies. Mouse infection experiments confirmed that bacterial densities in several organs steeply rise at later stages of infection (e.g., by about 100-fold between day 2 and day 5 [92]) and the formation of microcolonies in the liver and spleen of infected mice [93,94], suggesting that this phenotype may be relevant during infection. Our results partially contrast with the more uniform YopE expression (measured as YopE-mCherry signal) that was previously shown for *Y. pseudotuberculosis* microcolonies exposed to innate immune system cells [93]. A possible reason for this discrepancy is the higher temporal resolution of the reporter system used in our study ($P_{yopE}$-*sfGFP-ssrA*, S1 Fig), which also does not interfere with secretion, as has been shown for fusions of full-length T3SS effectors to fluorescent proteins [95]. The complete downregulation of expression and assembly of the T3SS components shown in our experiments further ensures that the T3SS cannot be unspecifically activated upon contact to eukaryotic cells during this later phase of the infection and especially during dissemination.

A key aspect of the $d^3T3$ is that besides all components of the T3SS, it also affects YadA, a trimeric autotransporter adhesin mediating the adhesion to extracellular matrix components of eukaryotic host cells [75,76,96] which is also part of the VirF regulon. As shown in Fig 7, this downregulation has a striking effect on bacterial attachment not only to each other, but also to eukaryotic cells, allowing bacteria to detach from eukaryotic cells, possibly leading to the disintegration of the microcolonies, and preventing reattachment during dissemination. YadA has a highly pleiotropic role during infection, leading to diverse outcomes across *Yersinia* species. It is essential for virulence in *Y. enterocolitica*, dispensable in *Y. pseudotuberculosis*, and detrimental if expressed in *Y. pestis*. In *Y. enterocolitica*, YadA contributes to serum resistance and exhibits a primarily antiphagocytic effect in synergy with the T3SS [75,76]. The antiphagocytic function of YadA is

largely attributed to its role in mediating tight adhesion of *Y. enterocolitica* to macrophages, facilitating T3SS activation and the injection of antiphagocytic effectors. Additionally, YadA promotes autoagglutination and microabscess formation, further hindering phagocytosis. The density-dependent downregulation of adhesion and type III secretion described in this manuscript is therefore likely to render the disseminating bacteria more vulnerable to phagocytosis. This raises the intriguing possibility that the density-dependent T3SS downregulation synchronizes bacterial dissemination as a strategy to optimize infection dynamics. Importantly, the reversibility of the d³T3 (Fig 3b) ensures that once bacteria are in a site with a lower local bacterial concentration, the T3SS can be reassembled, a process that takes 30–60 minutes [44], sufficient to disseminate in or between hosts, but ensuring that the T3SS is ready for the establishment of a fresh infection cycle at the new site.

In summary, we show that contrary to what was previously believed, *Y. enterocolitica* downregulates the expression of its T3SS and the associated adhesin YadA at higher bacterial densities, such as inside microcolonies. This specific downregulation may be conveyed by the CsrABC system via the main T3SS transcription factor VirF: Increased levels of the regulatory RNAs CsrB and especially CsrC at higher local densities sequester CsrA, which in the highly similar *Y. pseudotuberculosis* system destabilizes the *virF* transcript [55]. The resulting T3SS- and YadA-negative population can multiply and disseminate, even in the presence of potential target cells, an important aspect of bacterial virulence (Fig 8). Future experiments may reveal whether similar mechanisms ensure efficient replication and dissemination of *Shigella* and other T3SS-utilizing pathogens.

## Materials and methods

### Bacterial strain generation and genetic constructs

A list of strains and plasmids used in this study can be found in S6 Table. The *Y. enterocolitica* strains used in this study are based on the *Y. enterocolitica* wild-type strain MRS40 or its derivate IML421asd (ΔHOPEMTasd). IML421asd lacks the major virulence effector proteins (YopH,O,P,E,M,T) and harbors a deletion in the aspartate-beta-semialdehyde dehydrogenase gene, making the strain auxotrophic for diaminopimelic acid (DAP) and therefore suitable for work in a biosafety class 1 environment [97].

### Bacterial cultivation, secretion assay, and total protein assays

*Y. enterocolitica* overnight cultures were grown on a shaking incubator at 28°C in brain heart infusion medium (BHI) supplemented with nalidixic acid (Nal, 35 µg/ml) and DAP (60 µg/ml), where required. Day cultures (BHI supplemented with 35 µg/ml Nal, 60 µg/ml DAP where required, 20 mM $MgCl_2$, 0.4% glycerol, and 5 mM EGTA for secreting conditions or 5 mM $CaCl_2$ for non-secreting conditions) were inoculated from stationary overnight cultures (which typically were at an $OD_{600}$ of 12–18) to an $OD_{600}$ of 0.1, 0.3, 0.7, 1.0, or 1.5 by directly adding the required amount of the overnight culture to fresh medium. To select for the maintenance of expression plasmids, 200 µg/ml ampicillin was added, where required. Day cultures were incubated at 28°C for 90 min after the inoculum. Expression of the *yop* regulon was then induced by a rapid temperature shift to 37°C. Where needed, protein expression from pBAD-His/B (Invitrogen) derivative plasmids was induced at the temperature shift by adding 0.01–0.2% L-arabinose, as described for the respective experiments. Unless specified otherwise, bacteria were incubated for 150–180 min after the shift to 37°C. For secretion assays, 2 ml of the culture supernatant was harvested by centrifugation (10 min, 21,000 *g*) 180 min after the shift to 37°. Proteins in the supernatant were precipitated with 10% trichloroacetic acid (TCA) at 4°C for 1–8 h. Precipitated proteins were collected by centrifugation (15 min, 21,000 *g*, 4°C). After washing with ice-cold acetone (1 ml), the pellets were resuspended in SDS-PAGE loading buffer and normalized to 0.3 OD units (ODu)/15 µl for total cell analysis, or 0.6 ODu/15 µl for secretion assays (1 ODu = 1 ml of culture at $OD_{600}$ of 1, ~5 x $10^8$ *Y. enterocolitica*). After resuspension, samples were incubated for 5 min at 99°C and 15 µl were applied to SDS-PAGE analysis. Protein separation was performed on 15% SDS-PAGE gels and protein sizes were determined using the BlueClassic Prestained Marker (Jena Biosciences) as standard. For

visualization, the gels were stained with FastGene-Q-stain (NipponGenetics). Secretion efficiency was determined in Fiji [98] by gel densitometry for the YopH band, compared to the wild-type strain. For immunoblots, the separated proteins were transferred to a nitrocellulose membrane. Primary mouse anti-GFP antibody (ThermoFisher Proteintech 66002–1-Ig, 1:4000) was used in combination with a secondary anti-mouse antibody conjugated to horseradish peroxidase (GE Healthcare NXA931, 1:5000). Primary rabbit anti-mCherry antibody (BioVision 5993 1:2000) was used in combination with a secondary anti-rabbit antibody conjugated to horseradish peroxidase (Sigma-Aldrich A8275 1:10 000). For visualization, Immobilon Forte chemiluminescence substrate (Sigma-Aldrich) was used in a LAS-4000 Luminescence Image Analyzer.

## Growth curve and promoter activity assays

For growth curves and measurements of T3SS activation via upregulation of the YopE promoter (P*yopE*-sfGFP::ssrA), bacteria were treated as described above. After 90 min at 28°C, bacterial cultures were aliquoted in quadruplicates into a 96-well microplate (120 µl/well). sfGFP fluorescence (excitation: 480/25 nm, emission: 535/28 nm) for P*yopE* activity assays and absorbance at 600 nm for growth curves were measured at 37°C over time in a Tecan Infinite 200 Pro photometer. The measured absorbance was converted into $OD_{600}$ based on a calibration performed using the same culture volume and microtiter plate [99]. For sfGFP::ssrA degradation kinetics measurement, bacteria were treated as described before. After 120 min at 37°C in secreting-inducing conditions (5 mM EGTA), T3SS induction was stopped by adding 10 mM $Ca^{2+}$. Bacterial cultures were aliquoted in quadruplicates into a 96-well microplate and sfGFP fluorescence was measured as indicated above. For *virF*(-123/+75)-sfGFP reporter assays, bacteria were treated as described before. 45 min prior to the shift to 37°, 0.2% arabinose were added. Bacterial cultures were aliquoted in triplicates into a 96-well microplate and sfGFP fluorescence and $OD_{600}$ were measured every 5 min and corrected for the blank (medium only). The $OD_{600}$-adjusted fluorescence was corrected for the negative control (empty plasmid) and normalized to the positive control (pBAD::sfGFP) at the same conditions ($OD_{in}$ = 0.1 or 1.5).

## Fluorescence microscopy

For fluorescence microscopy, bacteria were grown as described above. After 2.5 h at 37°C, 500 µl of culture were harvested by centrifugation (2 min, 2400 *g*) and resuspended in 250 µl of microscopy medium (100 mM 2-[4-(2-Hydroxyethyl) piperazin-1-yl]ethane-1-sulfonic acid (HEPES) pH 7.2, 5 mM $(NH_4)_2SO_4$, 100 mM NaCl, 20 mM sodium glutamate, 10 mM $MgCl_2$, 5 mM $K_2SO_4$, 50 mM 2-(N-morpholino) ethane sulfonic acid (MES), 50 mM glycine). 2 µl of bacterial resuspension were spotted on an agarose pad (1.5% low melting agarose (Sigma-Aldrich) in microscopy medium, 1% casamino acids, 5 mM EGTA) in a glass depression slide (Marienfeld). For imaging, a Deltavision Elite Optical Sectioning Microscope equipped with a UPlanSApo 100 × /1.40 oil objective (Olympus) and an EDGE sCMOS_5.5 camera (Photometrics) was used. z stacks with 9 slices (Δz = 0.15 µm) were acquired. The GFP signal was visualized using a FITC filter set (excitation: 475/28 nm, emission: 535/48 nm) with 0.2 s exposure time. The mCherry signal was visualized using a mCherry filter set (excitation: 575/25 nm, emission: 625/45 nm) with 0.2 s exposure time. Images were processed with FIJI (ImageJ 1.51f/1.52i/1.52n) [98]. Fluorescence quantification was performed in FIJI. For visualization purposes, selected fields of view adjusted identically for brightness and contrast within the compared image sets are shown.

## Quantification of fluorescent foci

To quantify the number of stable EGFP-SctQ foci, individual *Y. enterocolitica* cells were first segmented using the neural network StarDist [100]. The training dataset was generated by manually annotating 17 images of *Y. enterocolitica* cells grown in BHI medium and imaged on agarose pads using the Deltavision Elite Optical Sectioning Microscope. To account for variation in the imaging plane, z-stacks with 0.15 µm spacing were recorded, of which 5 slices were selected for network training. The resulting 85 images were quartered, resulting in a final dataset of 300 (training) and 40 (validation) image pairs. Dataset size was increased 4-fold during training using flipping and rotation. The StarDist model was trained

for 100 epochs via the ZeroCostDL4Mic platform [101] using a batch size of 4, 260 steps, 120 rays, grid size of 2, an initial learning rate of 0.0003 and an train/test split of 80%/20%. The dataset and model are included in the DeepBacs [102] collection on Zenodo (doi: 10.5281/zenodo.11105050). Foci localization was performed using the Fiji plugin ThunderSTORM [103]. Localizations were identified using 3rd order B-Spline filtering and local maximum approximation with a peak intensity threshold of 4-fold standard deviation of the 1st wavelet level. For sub-pixel localization, peaks were fitted with an integrated Gaussian in a fitting radius of 3 pixels. Finally, localizations were filtered by PSF width (sigma < 600 nm) and images were rendered as a scatter plot with a magnification of 1. This allows for foci counting by measuring the integrated intensity within the ROIs of segmented cells. To ensure that the foci are located within the cell outlines, each ROI was enlarged by 2 pixels with the help of a custom-written Fiji macro. A manual of the entire procedure is provided as S1 File.

### Confocal microscopy imaging

For imaging of *Y. enterocolitica* microcolonies, 1 µl of a stationary overnight culture was spotted on an LB plate supplemented with NaI (35 µg/ml), DAP (60 µg/ml), glycerol (0.4%), and incubated at 28°C. For additional VirF expression, plates were supplemented with ampicillin (200 µg/ml) and Arabinose (0.2%) to ensure maintenance and induction of the expression plasmid. After 16 h, the plate was shifted to 37°C for 4 h. For colony imaging, colonies were deposited upside down in 8-well glass-bottom slides (µ-Slide, 8-well glass bottom; Ibidi, Gräfelfing), embedded in 250 µl of 1% low melting agarose (Sigma-Aldrich (Taufkirchen)). Confocal z-stacks were acquired on a Zeiss LSM-880 microscope, sfGFP-expressing cells were excited using an argon laser at 488 nm, through an LD-LCI Plan Apochromat 25x 0.8 NA multi-immersion objective, using water as immersion liquid. Fluorescence emission was collected in the 490–579 nm range. The pixel size (0.15 µm) and the distance between optical slices (1.18 µm) were adjusted according to the Nyquist criterion.

### Proteome analysis by shotgun proteomics-based mass spectrometry

Strains were grown and prepared as described above. After 2.5 h at 37°C, cultures were normalized to an $OD_{600}$ of 0.5, and 2 ml were harvested by centrifugation at 9,391 $g$ for 2 min. The cells were washed three times with ice-cold phosphate-buffered saline (PBS) (8 g/l NaCl, 0.2 g/l KCl, 1.78 g/l $Na_2HPO_4$*2H2O, 0.24 g/l $KH_2PO_4$, pH 7.4) (15,000 $g$, 10 min, 4°C) and resuspended in 300 µl lysis buffer (2% sodium lauroyl sarcosinate (SLS), 100 mM ammonium bicarbonate). Then samples were heated for 10 min at 90°C and sonicated with a vial tweeter after heating. Proteins were reduced with 5 mM Tris(2-carboxyethyl) phosphine (Thermo Fischer Scientific) at 90°C for 15 min and alkylated using 10 mM iodoacetamide (Sigma Aldrich) for 30 min at 20°C in the dark. Proteins were precipitated with a 6-fold excess of ice cold acetone and incubation for 2h at -20°C, followed by two methanol washing steps. Dried proteins were reconstituted in 0.2% SLS and the amount of proteins was determined by bicinchoninic acid protein assay (Thermo Scientific). For tryptic digestion, 50 µg protein was incubated in 0.5% SLS and 1 µg of trypsin (Serva) at 30°C over night. After digestion, SLS was precipitated by adding a final concentration of 1.5% trifluoroacetic acid (TFA, Thermo Fisher Scientific). Peptides were desalted by using C18 solid phase extraction cartridges (Macherey-Nagel). Cartridges were prepared by adding acetonitrile (ACN), followed by equilibration with 0.1% TFA. Peptides were loaded on equilibrated cartridges, washed with 5% ACN and 0.1% TFA containing buffer and finally eluted with 50% ACN and 0.1% TFA. Dried peptides were reconstituted in 0.1% Trifluoroacetic acid and then analyzed using liquid-chromatography-mass spectrometry carried out on an Exploris 480 instrument connected to an Ultimate 3000 RSLC nano and a nanospray flex ion source (all Thermo Scientific). The following separating gradient was used: 94% solvent A (0.15% formic acid) and 6% solvent B (99.85% acetonitrile, 0.15% formic acid) to 25% solvent B over 40 min, and an additional increase of solvent B to 35% for 20 min at a flow rate of 300 nl/min. MS raw data was acquired on an Exploris 480 (Thermo Scientific) in data independent acquisition (DIA) mode with a method adopted from [104]. The funnel RF level was set to 40. For DIA experiments full MS resolutions were set to 120.000 at m/z 200 and full MS, AGC (Automatic Gain Control) target was 300% with an IT of 50 ms. Mass range was set to 350–1400. AGC target value for fragment spectra was set at 3000%. 45 windows of 14 Da were used with an overlap of 1 Da. Resolution was set to 15,000 and IT to 22 ms. Stepped

HCD collision energy of 25, 27.5, 30% was used. MS1 data was acquired in profile, MS2 DIA data in centroid mode. Analysis of DIA data was performed using the DIA-NN version 1.8 [105] using a *Y. enterocolitica* GenBank protein database (https://www.ncbi.nlm.nih.gov/nuccore/NZ_CP011286.1) to generate a dataset-specific spectral library for the DIA analysis. The neural network based DIA-NN suite performed noise interference correction (mass correction, RT prediction and precursor/fragment co-elution correlation) and peptide precursor signal extraction of the DIA-NN raw data. The following parameters were used: Full tryptic digest was allowed with two missed cleavage sites, and oxidized methionine and carbamidomethylated cysteine residues. Match between runs and remove likely interferences were enabled. The precursor FDR was set to 1%. The neural network classifier was set to the single-pass mode, and protein inference was based on genes. Quantification strategy was set to any LC (high accuracy). Cross-run normalization was set to RT-dependent. Library generation was set to smart profiling. DIA-NN outputs were further evaluated using the SafeQuant [106,107] script modified to process DIA-NN outputs. The mass spectrometry proteomics data have been deposited to the ProteomeXchange Consortium via the PRIDE partner repository with the dataset identifier PXD052013reviewer_pxd052013@ebi.ac.uk).

### Sequence analysis

The CsrB and CsrC genetic regions of *Y. enterocolitica* KNG22703 (GenBank: CP011286.1) and *Y. pseudotuberculosis* YPIII (NZ_CP009792.1) [108] were compared. Alignments based on nucleotide sequence were performed by Clustal Omega.

### RT-qPCR

Total RNA from *Y. enterocolitica* cells grown as described above was extracted using the Monarch Total RNA Miniprep Kit (New England Biolabs). A $10^9$ cells pellet was first washed with 1 ml cold resuspension buffer (10 mM Tris-HCl pH 8.0, 100 mM NaCl, 1 mM EDTA) and resuspended in 250 µl cold resuspension buffer. 250 µl of lysis buffer (50 mM Tris/HCl pH 8.0, 8% Sucrose, 1% Triton X-100, 10 mM EDTA, 0.4% lysozyme) and 800 µl of Monarch Total RNA Miniprep Kit protectant buffer were added to the resuspension and incubated for 30 min at 4°C. After incubation, the cell suspension was incubated at 72°C for 15 min. RNA purification was performed according to the manufacturer's protocol. DNase treatment was performed using Turbo DNase (Thermo Fisher Scientific), followed by enzyme removal via Monarch RNA Cleanup Kit (New England Biolabs). cDNA was generated from 1 µg RNA using LunaScript RT SuperMix Kit (New England Biolabs). qPCR reactions were performed on an Applied Biosystems 7500 Real-Time PCR system using the Luna Universal qPCR MasterMix (New England Biolabs) and the primers listed in S7 Table. The DNA gyrase subunit *gyrB* was used as an internal control [109]. Data analysis was performed using the comparative threshold cycle ($C_T$) method [110].

### Attachment assay

In preparation for the cell adhesion assay, HeLa cells were seeded and grown overnight in individual wells of 24-well cell culture plates (Grainer). Cells were incubated with fresh RPMI medium supplemented with DAP (60 µg/ml) before the addition of bacteria. After 2h in secretion-inducing conditions (37°C, 5 mM EGTA), approximately 5 x $10^6$ bacteria were added to the PBS-washed HeLa monolayer (multiplicity of infection ~20) and incubated for 45 min. The attached cells were washed three times after incubation. HeLa cells and attached bacteria were collected via trypsinization and plated on *Yersinia* selective plates containing nalidixic acid (35 µg/ml) and DAP (60 µg/ml). After 24 h of incubation at 28°C, the number of colony-forming units was calculated per each condition tested.

### Supporting information

**S1 Fig. Kinetics of the P$_{yopE}$::*sfGFP-ssrA* activity assay.** 150 min after the induction of T3SS assembly by temperature shift to 37° in secreting conditions, T3SS activation in a wild-type strain carrying the P$_{yopE}$::*sfGFP-ssrA* reporter on the virulence plasmid was inhibited by adding 10 mM CaCl$_2$ to the medium (t = 0). From t = 0, fluorescence was measured over time. *n* = 3, graph shows representative result.
(PDF)

**S2 Fig. P$_{yopE}$::*sfGFP-ssrA* expression in *Y. enterocolitica* microcolonies.** Image of confocal microscopy sections (z = 0 µm, 13 µm, 27 µm) of a *Y. enterocolitica* Δ*sctW* P$_{yopE}$-*sfGFP-ssrA* microcolony section at 37°. T3SS activity, which results in a strong upregulation of the *yopE* promoter and cellular fluorescence, is only detected at the edge of the microcolony. Scale bar, 50 µm; *n* = 3.
(PDF)

**S3 Fig. *Y. enterocolitica* reference growth curve.** Optical density at 600 nm (OD$_{600}$) of *Y. enterocolitica* wild-type cultures (MRS40) incubated at 28°C. *n* = 3; whiskers denote standard deviation.
(PDF)

**S4 Fig. T3SS activity is repressed over time in growing cultures.** T3SS reporter assay (P$_{yopE}$-*sfGFP-SsrA*) of *Yersinia* cells inoculated at OD$_{in}$=0.1 in secreting medium. Culture aliquots were shifted to 37°C to induce assembly of the T3SS at different time points post-inoculum as indicated. T3SS activity was measured at a single-cell level 2.5 h post-induction. *n* = 3 independent experiments; each dot represents a single-cell measurement and the black bar represents the average of the fluorescence intensity.
(PDF)

**S5 Fig. *Y. enterocolitica* growth curves at different OD$_{in}$.** OD$_{600}$ of wild-type strain expressing P$_{yopE}$::*sfGFP-ssrA* under secreting conditions at the different OD$_{in}$ used in Fig 2b (0.1, 0.3, 0.7, 1.0, 1.5). OD$_{600}$ was measured from the time of the shift to 37° (t = 0). *n* = 3, whiskers denote standard deviation.
(PDF)

**S6 Fig. All T3SS gene categories are similarly downregulated at higher cell densities.** Volcano plot showing differences in the expression of *Y. enterocolitica* proteins between cultures grown under secreting conditions at OD$_{in}$ 1.5 and 0.1. All proteins with at least three detected peptides are displayed. Representation of the plot shown in Fig 3a (Table 1, S2 Table), highlighting different classes of proteins, colored according to their function.
(PDF)

**S7 Fig. The density-dependent downregulation of the T3SS is not conferred by quorum sensing.** Gel band intensity quantification of secretion assay of wild-type (WT, light blue) and quorum sensing deletion strain (ΔQS = Δ*yenI*, Δ*lsrK*, dotted bars) at OD$_{in}$ = 0.1, 0.3, 0.7, 1.0, 1.5 shown in Fig 4a. *n* = 4, each spot represents a single measurement, whiskers denote standard deviation. For statistics, WT and ΔQS measurements were compared for each OD$_{in}$ using an unpaired t-test. Each comparison resulted in a statistically non-significant difference (ns, *p* > 0.05), represented in the graph as grouped result.
(PDF)

**S8 Fig. The stringent response regulators RelA and SpoT do not significantly contribute to the density-dependent downregulation of the T3SS.** T3SS reporter assay (P$_{yopE}$-*sfGFP-SsrA*) of *Yersinia* cells at OD$_{in}$=0.1 and 1.5 in the indicated strains after shifting the culture to 37°C (t = 0), which induces the expression of the T3SS. *n* = 3, shadowed area denotes standard deviation.
(PDF)

**S9 Fig. VirF is essential for T3SS secretion.** a) Schematic display of the genes encoding for T3SS structural components (operons *virA*, *virB*, *virC*, and translocator operon (transl.), dark grey), T3SS effectors and chaperones (lighter shades of grey), and VirF (purple) on the pYV virulence plasmid (left), and the principle working mechanism of VirF (right). b) Secretion assay displaying the effectors secreted by the T3SS at OD$_{in}$ = 0.1 in wild-type (WT) and Δ*virF* background. Lanes 2–6, increasing expression of VirF *in trans* (arabinose concentrations of 0, 0.001, 0.003, 0.008%, respectively). *n* = 3, gel shows representative result.
(PDF)

**S10 Fig. Additional expression of the transcriptional regulator VirF restores T3SS secretion in a *Yersinia* micro-colony.** Image of confocal microscopy sections (z = 0 μm, 13 μm, 27 μm) of a *Y. enterocolitica* Δ*sctW* P*yopE*-*sfGFP-ssrA* microcolony additionally expressing VirF (induced by 0.2% arabinose) at 37°C. T3SS activity, visualized by the sfGFP signal, is detected throughout the colony. Scale bar, 50 μm, *n* = 3.
(PDF)

**S11 Fig. SctG protein levels are stable at higher local cell densities.** Expression of mCherry-SctL (expected molecular weight 51.4 kDa) and SctG-sfGFP (expected molecular weight 41.5 kDa) expressed from the native genetic location in *Y. enterocolitica* in secreting medium at the $OD_{in}$ indicated. Immunoblot using antibodies directed against mCherry and EGFP as indicated. Control: Expression of mCherry (expected size 26.7 kDa) and GFP (expected size 27.0 kDa) from plasmid. *n* = 3, blots show representative results.
(PDF)

**S12 Fig. Sequence conservation of CsrB and CsrC in *Y. enterocolitica* and *Y. pseudo-tuberculosis*.** DNA sequence alignment using Clustal Omega. Nucleotides that are identical in both sequences are indicated by asterisks.
(PDF)

**S13 Fig. A mutation in the CsrA binding region of *virF* reduces *virF* expression at low densities.** Relative fluorescence of a pBAD::*virF(-123/ +75)-sfGFP* reporter strain at $OD_{in}$ 0.1, compared to an identically treated pBAD::*sfGFP* control, measured in technical triplicates in a 96-well-plate after shifting the cultures to 37°C (t = 0), which induces the expression of the T3SS. Fluorescence is reduced in a (-16/-14: GGA to TTC) mutant (orange) with reduced CsrA binding [55] and at higher densities ($OD_{in}$ 1.5, open circles). *n* = 3 biological replicates.
(PDF)

**S14 Fig. Manipulations of the CsrABC system are efficiently compensated.** a) Changes in the transcript levels of CsrB and CsrC at $OD_{in}$ =1.5 upon CsrA overexpression (CsrA⁺, 0.2% arabinose) in comparison to the wildtype strain (WT), as measured by reverse transcription quantitative PCR. Additional CsrA expression is compensated by higher levels of CsrB and, to a lower degree, CsrC. b) T3SS reporter assay (P*yopE*-*sfGFP-ssrA*) at $OD_{in}$ 0.1 and 1.5 in the indicated strains after shifting the culture to 37°C (t = 0), which induces the expression of the T3SS. *n* = 3 for all panels; shadowed area in b) denotes standard deviation.
(PDF)

**S1 Table. Quantification of EGFP-SctQ foci per bacterium at different bacterial densities.** Quantification of EGFP-SctQ foci per bacterium, corresponding to assembled injectisomes, visualized by fluorescence microscopy at the indicated $OD_{in}$ values. Corresponds to Fig 2c.
(PDF)

**S2 Table. Density-dependent regulation of expression of non-T3SS components encoded on the *Yersinia* virulence plasmid.** While the adhesin YadA is downregulated similarly to the T3SS components (Table 1), most non-T3SS proteins encoded on the *Yersinia* pYV virulence plasmid are not strongly affected by the density-dependent downregulation. Label-free quantitative mass spectrometry in the total proteome of a ΔHOPEMTasd wild-type strain at the different growth conditions indicated, experiment and display format as shown in Table 1.
(PDF)

**S3 Table. The density-dependent downregulation of the T3SS differs from stationary phase response.** Label-free quantitative mass spectrometry of the known stationary response proteins CpxP, IclR and OsmY [a–d] in the total proteome of different growth conditions, as indicated. Stat., stationary cultures. # pept., number of total detected peptides. Display format as shown in Table 1 and S2 Table.
(PDF)

**S4 Table. CsrA levels are not not significantly altered at different densities.** Label-free quantitative mass spectrometry quantification of CsrA in the total proteome of a ΔHOPEMTasd wild-type strain at the different growth conditions indicated, experiment and display format as shown in Table 1.
(PDF)

**S5 Table. Density-dependent regulation of expression of proteins with CsrA-dependent transcription in *Y. pseudotuberculosis*.** Expression of genes affected by the absence of CsrA in *Y. pseudotuberculosis* [e]. All genes with gene identifier and with a > 4-fold transcription change in the *csrA* mutant were obtained from the data in the "Yersiniomics database" [f] (https://yersiniomics.pasteur.fr). Flagellar genes, which are not expressed at 37°, were excluded. Label-free quantitative mass spectrometry in the total proteome of a ΔHOPEMTasd wild-type strain at the different growth conditions indicated, experiment and display format as shown in Table 1.
(PDF)

**S6 Table. Strains and plasmids used in this study.**
(PDF)

**S7 Table. Oligonucleotides used in this study.** Nucleotide sequence of oligonucleotides used for the construction of plasmids or quantitative PCR (qPCR).
(PDF)

**S1 File. Foci quantification with StarDist and ThunderSTORM.**
(PDF)

## Author contributions

**Conceptualization:** Francesca Ermoli, Andreas Diepold.

**Data curation:** Francesca Ermoli, Gabriele Malengo, Christoph Spahn, Corentin Brianceau, Timo Glatter, Andreas Diepold.

**Formal analysis:** Timo Glatter.

**Funding acquisition:** Andreas Diepold.

**Investigation:** Francesca Ermoli, Christoph Spahn, Corentin Brianceau, Andreas Diepold.

**Methodology:** Francesca Ermoli, Gabriele Malengo, Christoph Spahn, Timo Glatter, Andreas Diepold.

**Project administration:** Andreas Diepold.

**Resources:** Timo Glatter, Andreas Diepold.

**Software:** Christoph Spahn.

**Supervision:** Andreas Diepold.

**Validation:** Timo Glatter, Andreas Diepold.

**Visualization:** Francesca Ermoli, Christoph Spahn, Timo Glatter, Andreas Diepold.

**Writing – original draft:** Francesca Ermoli, Andreas Diepold.

**Writing – review & editing:** Francesca Ermoli, Gabriele Malengo, Christoph Spahn, Corentin Brianceau, Andreas Diepold.

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
