## [Decision Letter · Decision Letter 0]

7 Mar 2025

Yersinia actively downregulates type III secretion and adhesion at higher cell densities

PLOS Pathogens

Dear Dr. Diepold,

Please submit your revised manuscript within 60 days May 06 2025 11:59PM. If you will need more time than this to complete your revisions, please reply to this message or contact the journal office at plospathogens@plos.org. Please include the following items when submitting your revised manuscript:

We look forward to receiving your revised manuscript.

Kind regards,

Jon Paczkowski

Academic Editor

PLOS Pathogens

David Skurnik

Section Editor

PLOS Pathogens

Editor-in-Chief

PLOS Pathogens

orcid.org/0000-0003-2946-9497

Editor-in-Chief

PLOS Pathogens

orcid.org/0000-0002-7699-2064

**Journal Requirements:**

At this stage, the following Authors/Authors require contributions: Francesca Ermoli, Gabriele Malengo, Christoph Spahn, Timo Glatter, and Andreas Diepold. Please ensure that the full contributions of each author are acknowledged in the "Add/Edit/Remove Authors" section of our submission form.

https://journals.plos.org/plospathogens/s/submission-guidelines#loc-parts-of-a-submission

Potential Copyright Issues:

- Figures 3C; Please confirm whether you drew the images / clip-art within the figure panels by hand. If you did not draw the images, please provide a link to the source of the images or icons and their license / terms of use; or written permission from the copyright holder to publish the images or icons under our CC BY 4.0 license. Alternatively, you may replace the images with open source alternatives. See these open source resources you may use to replace images / clip-art:

- The following Figure contains screenshots: Supplementary File 1. We are not permitted to publish these under our CC-BY 4.0 license, websites are usually intellectual property and are copyrighted.This includes peripheral graphics of the web browser such as icons and button. We ask that you please remove or replace it.

6) Please ensure that the funders and grant numbers match between the Financial Disclosure field and the Funding Information tab in your submission form. Note that the funders must be provided in the same order in both places as well.

**Reviewers' Comments:**

Reviewer's Responses to Questions

**Part I - Summary**

Reviewer #1: Ermoli et al. substantially revised their manuscript and appropriately addressed my concerns. Their new finding that T3SS is confined to less dense parts of a microcolony is particularly exciting and supports the author’s model. I congratulate the authors on this impressive work and recommend publication of the manuscript.

Reviewer #2: The authors show that expression of the Yersinia enterocolitica type 3 secretion system (T3SS) is downregulated at higher cell densities. This is an interesting phenomenon that may contribute to Y. enterocolitica pathogenesis. The authors provide compelling evidence that this regulation involves control of the level/activity of VirF, a transcription activator of T3SS genes. The authors argue that VirF levels are themselves regulated by the riboregulatory CsrA/B/C system. However, their evidence that the Csr system is responsible for the observed T3SS regulation is circumstantial and not well supported by the data.

Reviewer #3: In the present resubmitted study by Ermoli et al., the authors addressed the major points raised by the previous reviewers.

The authors convincingly showed that the Yersinia T3SS is downregulated/repressed during stationary phase and that this is not controlled by RpoS and the quorum sensing factors YenI and LsrK. In their resubmission they added that stationary phase control of T3SS is also not regulated by the stringent response RelA/SpoT. Moreover, they investigated T3SS activation in microcolonies in vitro using an sfGFP-ssrA fusion with high sensitivity and the potential for dynamic expression analyses due to the SsrA degradation tag, and found that indeed yopE expression is only induced in the growing/replicating bacteria at the surface of the colony.

The authors alternatively tested the influence of the post-transcriptional carbon starvation regulatory system CsrABC, as it was previously published to (i) influence Yersinia T3SS regulation via the control of VirF/LcrF, and (ii) vary in their expression during stationary phase in Yersinia. Based on these results and their observation that higher bacterial densities lead to higher CsrB and CsrC RNA levels, but not CsrA levels, they propose a model on CsrABC-mediated downregulation of T3SS expression.

**Part II – Major Issues: Key Experiments Required for Acceptance**

Reviewer #1: NA

Reviewer #2: 1. The authors conclude that virF expression is regulated as a function of cell density due to CsrA binding to the mRNA. However, the evidence for the involvement of CsrA is circumstantial. Figure S13B appears to address this question, but the figure panel is not mentioned in the text, and the graph is very difficult to interpret since the lines are difficult to tell apart. Mutating the CsrA binding sites in a virF translational reporter fusion would provide a more direct test of CsrA’s role.

2. CsrA regulates a large number of genes. If high cell density modulates CsrA activity, you would expect to see changes in the expression of other CsrA-regulated genes. The volcano plot (Figure 3A) suggests this is not the case, arguing against a model for T3SS regulation by cell density that involves CsrA. Nonetheless, it would be informative to reports changes in protein levels for known CsrA targets.

Reviewer #3: The results were obtained with quantitative fluorescence microscopy, comparative proteomic and qRT-PCR analysis etc. and have a very high quality, the density-mediated downregulation of T3SS expression is analyzed in detail and the date are very convincing.

However, the molecular mechanism how high density is sensed and transmitted to the known CsrABC-VirF-T3SS cascade and the relevance for virulence/pathogenesis or bacterial fitness remain unclear.

**Part III – Minor Issues: Editorial and Data Presentation Modifications**

Reviewer #1: NA

Reviewer #2: 1. Figure 3A. It would be helpful to indicate VirF on the volcano plot.

2. Line 209. Overexpressing virF restores T3SS expression at higher cell densities. The authors conclude that T3SS repression at higher cell densities is because of reduced virF expression, but it could be due to reduced VirF activity or activity of a repressive factor that counteracts the effects of VirF.

3. Line 230. “Indicate” is too strong a conclusion here. “Suggest” would be more appropriate. VirF translation or protein stability could be the regulated process, or there could be a promoter immediately upstream of virF.

4. Line 228. While SctG levels are less affected than those of VirF, it would be helpful for the authors to mention here that SctG levels are lower at the higher cell density.

5. Line 231-2. “To discriminate between transcriptional and post-transcriptional regulation”. Measuring RNA level does not distinguish between regulation of transcription and regulation of RNA stability.

Reviewer #3: lines 155-159

This interpretation is unclear to me “secretion was neither suppressed at low density by spent medium”.

In Fig. 3c the fluorescence of the PyopE-sfgfp-ssrA reporter is decreased (P<0.0001) after media change. Please explain.

PLOS authors have the option to publish the peer review history of their article (what does this mean? ). If published, this will include your full peer review and any attached files.

**Do you want your identity to be public for this peer review?** For information about this choice, including consent withdrawal, please see our Privacy Policy .

Reviewer #1: No

Reviewer #2: No

Reviewer #3: No

**Figure resubmission:**

**Reproducibility:**



---

## [Decision Letter · Decision Letter 1]

2 Jul 2025

Yersinia actively downregulates type III secretion and adhesion at higher cell densities

PLOS Pathogens

Dear Dr. Diepold,

Please submit your revised manuscript within 60 days Aug 31 2025 11:59PM. If you will need more time than this to complete your revisions, please reply to this message or contact the journal office at plospathogens@plos.org. Please include the following items when submitting your revised manuscript:

We look forward to receiving your revised manuscript.

Kind regards,

Jon Paczkowski

Academic Editor

PLOS Pathogens

David Skurnik

Section Editor

PLOS Pathogens

Editor-in-Chief

PLOS Pathogens

orcid.org/0000-0003-2946-9497

Editor-in-Chief

PLOS Pathogens

orcid.org/0000-0002-7699-2064

**Journal Requirements:**

Please ensure that the funders and grant numbers match between the Financial Disclosure field and the Funding Information tab in your submission form. Note that the funders must be provided in the same order in both places as well.

**Reviewers' Comments:**

Reviewer's Responses to Questions

**Part I - Summary**

Reviewer #2: My main concern with the original submission was the lack of compelling evidence to support the idea that the effect of cell density on T3SS gene expression was due to CsrA. The authors have made two changes to the manuscript to address this question. First, they show that protein levels for other CsrA-regulated genes are affected much less by cell density than protein levels for T3SS components. They argue that the effects of CsrA are lower when the T3SS is expressed, but the effects of CsrA on the T3SS would presumably also be lower if that were the case. Second, the authors made a virF translational reporter construct, and looked at the effect of mutating the CsrA binding site. Their data support the idea that CsrA regulates virF expression. If their model is correct, you would expect that the expression of the mutant reporter fusion would be unaffected by cell density, but they omit this key experiment. Moreover, the relatively small effect of mutating the CsrA binding site compared to the effect of cell density argues against a role for CsrA in the cell density-dependent regulation. Similarly, deleting csrC has little impact on density-dependent regulation, although the authors argue that autoregulation in the Csr system may buffer the effect of deleting csrC. Overall, I remain unconvinced of the role of CsrA in the density-dependent regulation.

**Part II – Major Issues: Key Experiments Required for Acceptance**

Reviewer #2: The authors should measure expression of the mutant reporter construct from Supplementary Figure 13 at ODin 1.5.

**Part III – Minor Issues: Editorial and Data Presentation Modifications**

Reviewer #2: Minor comment: the authors should describe Supplementary Figure 14 and their interpretation of these data in the Results section.

PLOS authors have the option to publish the peer review history of their article (what does this mean? ). If published, this will include your full peer review and any attached files.

**Do you want your identity to be public for this peer review?** For information about this choice, including consent withdrawal, please see our Privacy Policy .

Reviewer #2: No

**Figure resubmission:**

**Reproducibility:**



---

## [Decision Letter · Decision Letter 2]

3 Aug 2025

Dear Prof. Dr. Diepold,

We are pleased to inform you that your manuscript 'Yersinia actively downregulates type III secretion and adhesion at higher cell densities' has been provisionally accepted for publication in PLOS Pathogens.

Best regards,

Jon Paczkowski

Academic Editor

PLOS Pathogens

David Skurnik

Section Editor

PLOS Pathogens

Sumita Bhaduri-McIntosh

Editor-in-Chief

PLOS Pathogens

orcid.org/0000-0003-2946-9497

Michael Malim

Editor-in-Chief

PLOS Pathogens

orcid.org/0000-0002-7699-2064

Reviewer Comments (if any, and for reference):

Reviewer's Responses to Questions

**Part I - Summary**

Reviewer #2: The reviewers have modified the text to soften their conclusions about the role of CsrA in density-dependent regulation. Overall, I think the paper describes a very interesting phenomenon. I recommend publication without further changes.

**Part II – Major Issues: Key Experiments Required for Acceptance**

Reviewer #2: (No Response)

**Part III – Minor Issues: Editorial and Data Presentation Modifications**

Reviewer #2: (No Response)

PLOS authors have the option to publish the peer review history of their article (what does this mean? ). If published, this will include your full peer review and any attached files.

**Do you want your identity to be public for this peer review?** For information about this choice, including consent withdrawal, please see our Privacy Policy .

Reviewer #2: No

---

## [Editor Report · Acceptance letter]

Dear Prof. Dr. Diepold,

We are delighted to inform you that your manuscript, " 

Yersinia actively downregulates type III secretion and adhesion at higher cell densities," has been formally accepted for publication in PLOS Pathogens.

Best regards,

Sumita Bhaduri-McIntosh

Editor-in-Chief

PLOS Pathogens

orcid.org/0000-0003-2946-9497

Michael Malim

Editor-in-Chief

PLOS Pathogens

orcid.org/0000-0002-7699-2064